



# Recent ozone trends in the Chinese free troposphere: role of the local emission reductions and meteorology

Gaëlle Dufour[1], Didier Hauglustaine[2], Yunjiang Zhang[2,*], Maxim Eremenko[3], Yann Cohen[2], Audrey Gaudel[4], Guillaume Siour[3], Mathieu Lachatre[5,**], Axel Bense[3], Bertrand Bessagnet[6,***], Juan Cuesta[3], Jerry Ziemke[7], Valérie Thouret[8], Bo Zheng[9]

[1] Université de Paris and Univ Paris Est Creteil, CNRS, LISA, F-75013 Paris, France
[2] Laboratoire des Sciences du Climat et de l'Environnement (LSCE), UMR 8212, CEA-CNRS-UVSQ, Gif-sur-Yvette, France
[3] Univ Paris Est Creteil and Université de Paris, CNRS, LISA, F-94010 Créteil, France
[4] CIRES, University of Colorado/NOAA Chemical Sciences Laboratory, Boulder, CO, USA
[5] LMD/IPSL, École Polytechnique, Institut Polytechnique de Paris, ENS, PSL Université, Sorbonne Université, CNRS, Palaiseau, France
[6] Ecole Polytechnique, Institut Polytechnique de Paris, ENS, PSL Universite, Sorbonne Universite, CNRS, 91128 Palaiseau, France
[7] NASA Goddard Space Flight Center, and Goddard Earth Sciences Technology and Research (GESTAR), Maryland, USA
[8] Laboratoire d'Aérologie, Université de Toulouse, CNRS, UPS, France
[9] Institute of Environment and Ecology, Tsinghua Shenzhen International Graduate School, Tsinghua University, Shenzhen 518055, China

* now at School of Environmental Science and Engineering, Nanjing University of Information Science and Technology, Nanjing 210044, China
** now at ARIA Technologies 8-10 rue de la Ferme 92100 Boulogne-Billancourt France
*** now at European Commission, Joint Research Centre (JRC), Ispra, Italy

*Correspondence to*: Gaëlle Dufour (gaelle.dufour@lisa.ipsl.fr)

## Abstract

Free tropospheric ozone ($O_3$) trends in the Central East China (CEC) and export regions are investigated for 2008-2017 using the IASI $O_3$ observations and the LMDZ-OR-INCA model simulations, including the most recent Chinese emission inventory. The observed and modeled trends in the CEC region are -0.07 ± 0.02 DU/yr and -0.08 ± 0.02 DU/yr respectively for the lower free troposphere (3-6km column), and -0.05 ± 0.02 DU/yr and -0.06 ± 0.02 DU/yr respectively for the upper free troposphere (6-9km column). The statistical p-value is smaller to 0.01 for all the derived trends. A good agreement between the observations and the model is also observed in the region including Korea and Japan and corresponding to the region of pollution export from China. Based on sensitivity studies conducted with the model, we evaluate at 60% and 52% the contribution of the Chinese anthropogenic emissions to the trend in the lower and upper free troposphere, respectively. The second main contribution to the trend is the meteorological variability (34% and 50% respectively). These results suggest that the reduction of $NO_x$ anthropogenic emissions that occurred since 2013 in China lead to a decrease in ozone in the Chinese free troposphere, contrary to the increase in ozone at the surface. We designed some tests to compare the trends derived by the IASI observations and the model to independent measurements such as IAGOS or other satellite measurements (OMI/MLS). These comparisons



do not confirm the O₃ decrease and stress the difficulty to analyze short-term trends using multiple datasets with various sampling and the risk to overinterpret the results.

## 1 Introduction

Tropospheric ozone is a harmful pollutant close to the surface impacting human health and ecosystems (Lelieveld et al., 2015; Monks et al., 2015). Tropospheric ozone is also a short-lived climate forcer with an impact on surface temperature greatest in the upper troposphere lower stratosphere (UTLS) and then contributes to climate change (Riese et al., 2012). The recent Tropospheric Ozone Assessment Report (TOAR) has stated that free tropospheric O₃ increased during industrial times and the last decades (Gaudel et al., 2018; Tarasick et al., 2019). At the surface, the trends depend on the considered regions: a decrease is observed during summertime in North America and in Europe, and an increase is observed in Asia (e.g. Gaudel et al., 2018, 2020). However, conclusions are more difficult to draw for the recent trends of tropospheric ozone. In addition to the statistical robustness of these trends, Gaudel at al. (2018) point out inconsistencies between satellite trends derived from ultraviolet (UV) sounders, which show mainly positive trends (e.g. Cooper et al., 2014; Ziemke et al., 2019) and infrared (IR) sounders, which shows mainly negative trends (Wespes et al., 2017).

In China and Central East China (CEC), one of the most polluted regions worldwide (e.g. Wang et al., 2017; Fan et al., 2020), stringent pollutant emission controls for NOₓ, SO₂, and primary PM (particulate matter) emissions have been applying during the last decade (Zhang et al., 2019; Zheng et al., 2018). The main objective of these restrictions was to decrease primary and secondary PM concentrations (e.g. Zhai et al., 2019; Zhang et al., 2019). However, these reductions have leaded to a worsening of urban ozone pollution (Li et al., 2020; Liu and Wang, 2020a, b; Lu et al., 2018; Ma et al., 2021), attributed to O₃-precursors reductions in the large urban VOC-limited regions and directly to the aerosol reductions, which slow down the aerosol sink of hydroperoxy radicals (RO₂) and then increase the ozone production (Li et al., 2019; Ma et al., 2021). Most of the studies are based on surface observations and model simulations.

Satellite observations are more difficult to use to derive information on surface ozone due to their lack of sensitivity to surface concentration. Shen et al. (2019) show a relatively good correlation between OMI and surface measurements, especially in Southern China and state a possibility to infer trends for the subtropical latitudes. This was already partly reported by Hayashida et al. (2015). For individual events, IASI (Dufour et al., 2015) and IASI+GOME2 (Cuesta et al., 2018) products show abilities to inform on pollution events in the North China Plain. The IASI+GOME2 O₃ product shows a better ability to reproduce ozone surface concentrations with good comparisons with surface measurements in Japan (Cuesta et al., 2018). Despite this encouraging partial sensitivity to surface or boundary layer ozone, satellite observations such as IASI are mostly suited to probe free tropospheric ozone. IASI is however able to separate, at least partly, the information from the lower and the upper troposphere with a maximum of sensitivity between 3 and 6 km (Dufour et al., 2010, 2012, 2015). Based on the IASI observations, Dufour et al. (2018) discuss lower tropospheric O₃ trends (surf-6km) over the NCP for the 2008-2016 period and associate driving factors using a multivariate regression model. They show that O₃ trend derived from IASI is negative (-0.24



DU/yr or -1.2 %/yr) and explained by large-scale dynamical processes such as El Niño and changes in precursors emissions since 2013. The hypothesis to explain the negative impact of precursors reduction compared to the positive one at the surface is related to the chemical regime turning from VOC-limited at the surface to $NO_x$-limited in altitude. In this study, we question the ability of IASI to derive free tropospheric ozone trends in China by comparison with the state-of-the-art global chemistry-

5 climate model LMDz-OR-INCA for the 2008-2017 period. Satellite observations and the model are evaluated using independent observations (surface measurements, IAGOS aircraft measurements and ozone sondes). We use the model to quantify independently from IASI the contributions of local anthropogenic emissions and other possible driving factors (meteorology, global anthropogenic emissions, biomass burning, methane). Results are also discussed in light with the TOAR outcomes. The domain and regions of interest of our study are shown in Fig 1. Section 2 provides a description of the different

satellite and in-situ data, and the CTM. The IASI ozone product and the model simulations are evaluated against independent in-situ measurements and compared in section 3. Section 4 presents the observed and simulated $O_3$ trends in the troposphere. The results are discussed in section 5.

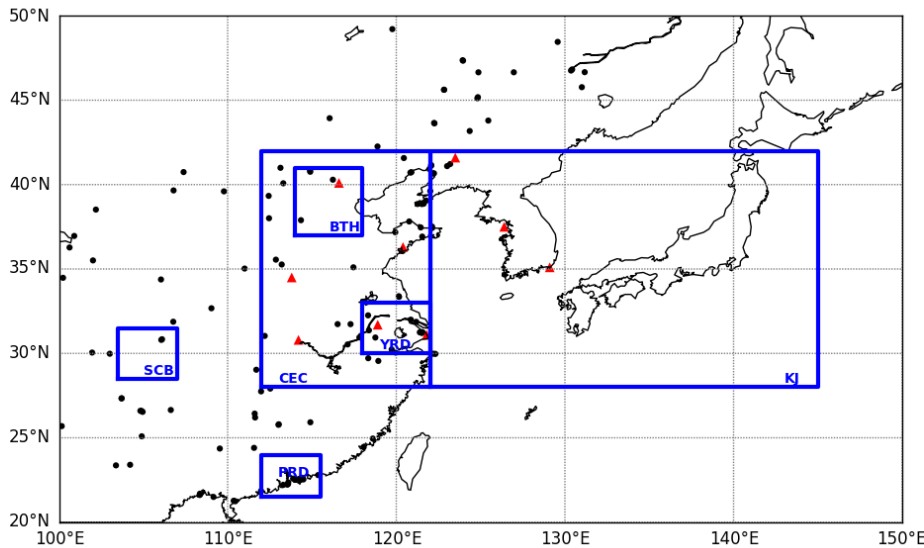

**Figure 1: Northeast Asian domain of the study. Six subregions of interest are considered: the Central East China (CEC, 28-42°N, 112-122°E), the Beijing-Tianjin-Hebei region (BTH, 37-41°N, 114-118°E), the Yangtze River Delta (YRD, 30-33°N, 118-122°E), the Pearl River Delta (PRD, 21.5-24°N, 112-115.5°E), the Sichuan Basin (SCB, 28.5-31.5°N, 103.5-107°E), and the Korea-Japan region (KJ, 28-42°N, 122-145°E). Black circles show the rural-type surface stations and the red triangles the IAGOS airports used in this study.**



## 2 Data and model description

### 2.1 IASI satellite data and ozone retrieval

The IASI (Infrared Atmospheric Sounding Interferometer) instruments are nadir-viewing Fourier transform spectrometers. They are flying on board the EUMETSAT (European Organisation for the Exploitation of Meteorological Satellites) Metop

satellites (Clerbaux et al., 2009). Three versions of the instrument are currently operational on the same orbit: one aboard the Metop-A platform since October 2006, one aboard the Metop-B platform since September 2012, and one aboard the Metop-C platform since November 2018. The IASI instruments operate in the thermal infrared between 645 and 2760 cm$^{-1}$ with an apodized resolution of 0.5 cm$^{-1}$. The field of view of the instrument is composed of a $2 \times 2$ matrix of pixels with a diameter at nadir of 12 km each. IASI scans the atmosphere with a swath width of 2200 km and crosses the equator at two fixed local solar

times 9:30 am (descending mode) and 9:30 pm (ascending mode), allowing the monitoring of atmospheric composition twice a day at any location.

Ozone profiles are retrieved from the IASI radiances using the KOPRA radiative transfer model, its inversion tool (KOPRAFIT) and an analytical altitude-dependent regularization method as described in (Eremenko et al., 2008) and (Dufour et al., 2012, 2015). In order to avoid the potential impact of versioning of the auxiliary parameters (such as temperature profile,

clouds screening, etc) on the ozone retrieval (Van Damme et al., 2017), surface temperature and temperature profiles are retrieved before the ozone retrieval. A data screening procedure is applied to filter cloudy scenes and to insure the data quality (Eremenko et al., 2008; Dufour et al., 2010, 2012). Three different a priori and constraints (polar < 10 km, midlatitudes – 10-14 km, tropical > 14 km) are used depending on the tropopause height, which is based on the 2 PV geopotential height product from the ECMWF (European Center for Medium-range Weather Forecasts). The a priori profiles are compiled from the

ozonesonde climatology of McPeters et al. (2007). Compared to the previous version of the ozone product (Dufour et al., 2018), water vapor is fitted simultaneously with ozone to account for remaining interferences in the spectral windows used for the retrieval and improve the retrieval in the current version v3.0 of the product. From the retrieved profiles, different ozone partial columns can be calculated. In this study, we consider four partial columns: the lowermost tropospheric (LMT) column from the surface to 3 km (named 0-3km), the lower free tropospheric (LFT) column from 3 km to 6 km (named 3-6km), the

upper free tropospheric (UFT) column from 6 km to 9 km (named 6-9km), and the upper tropospheric – lowermost stratospheric (UT-LMS) column from 9 km to 12 km (named 9-12km). Note that only the morning overpasses of IASI are considered for this study in order to remain in thermal conditions with a better sensitivity to the lower troposphere. To cover a larger period, we also consider only IASI on Metop-A.

### 2.2 Ozonesondes

Ozonesondes measure in situ vertical profiles of temperature, pressure, humidity and ozone up to 30–35 km with a vertical resolution of ~150 m for ozone. The ozonesondes data come mainly from the WOUDC database (http://www.woudc.org/), and the SHADOZ database (http://croc.gsfc.nasa.gov/shadoz/). The sonde measurements use electrochemical concentration





cell (ECC) technique, relying on the oxidation of ozone with a potassium iodine (KI) solution (Komhyr et al., 1995), except the Hohenpeissenberg sondes, which use Brewer Mast type sondes. Their accuracy for the ozone concentration measurement is about ±5 % (Deshler et al., 2008; Smit et al., 2007; Thompson et al., 2003). We use a database of ozonesonde measurements from 2007 to 2012 including 24 stations in the midlatitudinal bands (30-60°, both hemispheres), 13 stations in the tropical

band (30°S-30°N) and 16 stations in the polar bands (60-90°, both hemispheres). A list of stations and related information is provided in Table A1 (Appendix A).

## 2.3 Surface measurements

Observational data are issued from a freely available dataset accessible at https://quotsoft.net/air/ (last access: 3 May 2021). The dataset provides hourly data of criteria pollutants $SO_2$, $O_3$, $NO_2$, CO, $PM_{2.5}$ and $PM_{10}$ consolidated every day in near real

time from May 2014. Only national level data are available in this dataset for about 1300 stations through mainland China. In this study we consider only the stations with more than 50% of measurements available to ensure a good temporal coverage for the entire period (2014-2017). In the domain shown in Fig. 1, this corresponds to 685 stations. We classified the stations in different types of environment: mountain, rural, suburban, urban and traffic based on the approach developed by Flemming et al. (2005) for Europe. This method has the advantage of not requiring any additional information other than the pollutant

concentration. The relative amplitude of the diurnal cycle of $O_3$ observations is used to evaluate the representative environment of the station: the larger the amplitude of the diurnal ozone cycle is, the more the station is in an environment close to anthropogenic sources. In our case, each station has been evaluated on the studied period (i.e. 2014-2017).

## 2.4 IAGOS data

IAGOS (In-Service Aircraft for Global Observing System, http://www.iagos.org) is a European Research Infrastructure

dedicated to measure air composition (Petzold et al., 2015). The program counts more than 62,000 flights between 1994 and 2021 with ozone measurements. For the purpose of this study, we used all profiles of ozone at any time of day available above Northeast China / Korea between 2011 and 2017. On board the IAGOS commercial aircraft, ozone is measured using dual-beam ultraviolet absorption monitor (time resolution of 4 s) with an accuracy and a precision estimated at about 2 nmol mol-1 and 2% respectively. Further information on the instrument is available in (Thouret et al., 1998; Nédélec et al., 2015). Long-

term quality and consistency have been assessed by Blot et al. (2020).

## 2.5 OMI/MLS

OMI/MLS tropospheric column ozone is described by Ziemke et al. (2019). The OMI/MLS ozone product represents monthly means for October 2004-present at 1° × 1.25° resolution and latitude range 60°S - 60°N. Tropospheric column ozone is determined by subtracting co-located Microwave Limb Sounder (MLS) stratospheric column ozone from OMI total column

ozone each day at each grid point. Tropopause pressure used to determine MLS stratospheric column ozone invoked the WMO



K.km$^{-1}$ lapse-rate definition from NCEP re-analyses. OMI total ozone data are available from https://ozonewatch.gsfc.nasa.gov/data/omi/ (last access: 22 April 2021). MLS ozone data can be obtained from https://mls.jpl.nasa.gov/products/o3_product.php/ (last access: 22 April 2021). Estimated 1σ precision for the OMI/MLS monthly-mean gridded TCO product is 1.3 DU.

## 2.6 LMDZ-OR-INCA model

The LMDZ-OR-INCA global chemistry-aerosol-climate model (hereafter referred to as INCA) couples on-line the LMDZ (Laboratoire de Météorologie Dynamique, version 6) General Circulation Model (Hourdin et al., 2006) and the INCA (INteraction with Chemistry and Aerosols, version 5) model (Hauglustaine et al., 2004). The interaction between the atmosphere and the land surface is ensured through the coupling of LMDZ with the ORCHIDEE (ORganizing Carbon and Hydrology In Dynamic Ecosystems, version 9) dynamical vegetation model (Krinner et al., 2005). In the present configuration, the model includes 39 hybrid vertical levels extending up to 70 km. The horizontal resolution is 1.25° in latitude and 2.5° in longitude. The primitive equations in the GCM are solved with a 3 min time-step, large-scale transport of tracers is carried out every 15 min, and physical and chemical processes are calculated at a 30 min time interval. For a more detailed description and an extended evaluation of the GCM we refer to Hourdin et al. (2006). INCA initially included a state-of-the-art CH$_4$-NO$_x$-CO-NMHC-O$_3$ tropospheric photochemistry(Hauglustaine et al., 2004; Folberth et al., 2006). The tropospheric photochemistry and aerosols scheme used in this model version is described through a total of 123 tracers including 22 tracers to represent aerosols. The model includes 234 homogeneous chemical reactions, 43 photolytic reactions and 30 heterogeneous reactions. Please refer to Hauglustaine et al. (2004) and Folberth et al. (2006) for the list of reactions included in the tropospheric chemistry scheme. The gas-phase version of the model has been extensively compared to observations in the lower-troposphere and in the upper-troposphere. For aerosols, the INCA model simulates the distribution of aerosols with anthropogenic sources such as sulfates, nitrates, black carbon, particulate organic matter, as well as natural aerosols such as sea-salt and dust. Ammonia and nitrates aerosols are considered as described by Hauglustaine et al. (2014). The model has been extended to include an interactive chemistry in the stratosphere and mesosphere (Terrenoire et al., 2021). Chemical species and reactions specific to the middle atmosphere were added to the model. A total of 31 species were added to the standard chemical scheme, mostly belonging to the chlorine and bromine chemistry, and 66 gas phase reactions and 26 photolytic reactions.

In this study, meteorological data from the European Center for Medium-Range Weather Forecasts (ECMWF) ERA-Interim reanalysis have been used to constrain the GCM meteorology and allow a comparison with measurements. The relaxation of the GCM winds towards ECMWF meteorology is performed by applying at each time step a correction term to the GCM u and v wind components with a relaxation time of 2.5 h (Hauglustaine et al., 2004). The ECMWF fields are provided every 6 hours and interpolated onto the LMDZ grid.

The historical global anthropogenic emissions are taken from the Community Emissions Data System (CEDS) inventories (Hoesly et al., 2018) up to 2014. After 2014, the global anthropogenic emissions are based on Gidden et al. (2019). For China,





the anthropogenic emission inventories are replaced by the Zheng et al. (2018) emissions available for the period 2010-2017. The global biomass burning emissions are taken from van Marle et al. (2017) up to 2015 and from Gidden et al. (2019) after 2015. The ORCHIDEE vegetation model has been used to calculate off-line the biogenic surface fluxes of isoprene, terpenes, acetone and methanol as well as NO soil emissions as described by Messina et al. (2016).

## 3 Evaluation of the IASI $O_3$ satellite product and model simulations

### 3.1 Validation of the IASI $O_3$ product with ozonesondes and IAGOS

We present here a short validation of the version v3.0 of the IASI $O_3$ product developed by LISA. A detailed validation will be provided in a dedicated paper. The coincidence criteria used for the validation are 1° around the station, a time difference smaller than +/-6 hours and a minimum of 10 clear-sky pixels matching these criteria. No correction factor has been applied on ozonesonde measurements as our main concern is the (lower) troposphere. The results of the comparison between IASI ozone retrievals and ozonesonde measurements are summarized in Table 1 for different partial columns in the troposphere. Normalized mean biases (NMB) for the different partial columns remain very small (<2%). The estimated errors given by the normalized root mean square error (NRMSE) range between 10 and 20% depending on the partial columns and the Pearson correlation coefficient (R) is larger or equal to 0.79. Note that these results are based on the comparison with ozonesonde profiles smoothed by the averaging kernels of the IASI retrieval. If we compare with the raw sonde profiles without any smoothing, the results are slightly degraded but remain good with the normalized biases within +/-5%, the errors smaller than 30% and the correlations larger than 0.6. Version v3.0 of the $O_3$ IASI product reduces biases and increases the correlation with the ozone sondes measurements. The bias reduction is the most effective in the upper troposphere.

**Table 1: Statistics of the comparison of different $O_3$ partial columns derived from IASI with $O_3$ partial columns measured with ozonesondes for 2007-2012 all the globe and with IAGOS for 2011-2017 for North China - Korea. The Normalized Mean Bias (NMB), the Normalized Root Mean Square Error (NRMSE), and the Pearson correlation coefficient (R) are provided.**

|  | Ozonesondes – 2007-2012 | | | IAGOS – 2011-2017 | | |
|---|---|---|---|---|---|---|
| Columns | NMB (%) | NRMSE(%) | R | NMB (%) | NRMSE (%) | R |
| Surf-3km | -0.1 | 9.8 | 0.92 | -2.5 | 14 | 0.45 |
| 3-6km | 0.5 | 14.8 | 0.79 | -1.8 | 17 | 0.62 |
| 6-9km | 1.6 | 18.9 | 0.84 | -2.5 | 25 | 0.59 |
| 9-12km | 1.3 | 18.6 | 0.92 | | | |
| Surf-6km | 0.2 | 12.3 | 0.84 | | | |
| Surf-12km | 0.9 | 13.7 | 0.89 | | | |

In addition, we compared IASI ozone partial columns with IAGOS ozone partial columns, calculated from profiles measured above Chinese and Korean airports for the period 2011-2017. In this region, the IAGOS coverage was too sparse before this period. We use coincidence criteria similar to those of the ozonesondes for the IASI pixels taking the latitude and longitude of





the center of the IAGOS profile as reference. One difficulty for comparing IASI and IAGOS arises from the top of the IAGOS profiles which are much lower in altitude than for the sondes for example. We select IAGOS profiles with top measurements not lower than 500 hPa: 213 profiles are then selected for 2011-2017 in coincidence with IASI over about 1000 IAGOS profiles available before filtering. We extend the IAGOS profiles with the a priori profiles used in the IASI retrieval from the top of

IAGOS measurements to 60 km altitude in order to apply the averaging kernels. Then, the comparison between IASI and IAGOS is the most meaningful below 500hPa and for the partial columns representative of the lower free troposphere. We focus more on the lower free tropospheric column (3-6km) where the IASI retrievals are the most sensitive (Dufour et al., 2012). Results are summarized in Table 1. The normalized mean bias and the normalized root mean square error estimate between IASI and IAGOS are -1.8% and 17% respectively when AK are applied to IAGOS profiles (-5.6% and 18% without

AK applied) for this column in agreement with global ozonesonde validation. The correlation is smaller (0.62) than the one with the sondes (0.79). Statistics for the other columns are also reported in Table 1. The agreement is still good in terms of bias in the LMT (0-3km) column but degrades in terms of correlation. The temporal coincident criterion (+/- 6 hours) combined with a non-negligible influence of the ozone diurnal cycle (Petetin et al., 2016) on the LMT (0-3km) column might explain the worser correlation for this column (0.45). For the 6-9km column, only 50 profiles over 213 reach 300 hPa, then the IAGOS

profiles are largely mixed with the a priori profile used in the IASI retrieval. The evaluation of IASI using IAGOS is then difficult for this column.

## 3.2 Evaluation of the INCA simulations

The model is evaluated using the Chinese surface network described in section 2.3. As the model resolution is coarse and not representative of urban situations, we compare the model only with the rural type stations. The daily $O_3$ concentrations

simulated by the model are compared to the daily averages calculated from the hourly surface measurements provided by the Chinese surface network. The normalized mean bias between INCA and the surface stations is 12% over the Chinese domain considered, INCA being larger. The correlation and the normalized root mean square error (NRMSE) are 0.42 and 50% respectively. On average, the model shows relatively good performances, especially in terms of bias. However, the performances of the model to reproduce the ozone concentration depend on the region. A good agreement is observed in the

CEC region with bias of 3%, correlation of 0.49, and NRMSE of 48% respectively (Table 2). In the BTH region, north of the CEC, the modeled $O_3$ concentrations are smaller than the observed ones (-13%) but the correlation is higher (0.68). In the south of the CEC, the comparison in the YRD region remains satisfying in terms of bias (18%) but is degraded for the correlation (0.34) and the NRMSE (53%). In the coastal region of PRD, the available stations are within one model grid cell including land and sea. The coarse resolution of the model likely limits its capability to reproduce correctly the $O_3$

concentrations of the coastal stations: the model overestimates the surface measurements (41%) with large NRMSE (60%) and poor correlation (0.44). In the SCB, too few stations are available to provide statistics for the comparison. Figure A1 (Appendix A) shows the comparison station by station. Similar results are shown with a very good agreement in the northern part of the domain: biases within 10%, correlation larger than 0.6 and NRMSE smaller than 40%. In the southern part of the domain, the





model has some difficulties to reproduce the observations with biases ranging from 30% to 60% for most of the stations, larger for some stations. The correlation is limited and the NRMSE is larger than 50%.

**Table 2. Comparison between the daily O₃ concentrations simulated by the INCA model and the daily mean O₃ concentrations**
**measured at the Chinese surface stations for 2014-2017. Due to the model resolution, only the stations classified as rural are used**
**for the comparison. For each region, the number of stations as well as the mean observed and simulated O₃ concentrations are**
**provided. The normalized mean bias (NMB in %) is calculated as the difference between the model concentration and the observed**
**one, the latter used as reference. The normalized root-mean-square error (NRMSE in %) is calculated based on the daily averages,**
**and the correlation coefficient (r) corresponds to the temporal and the spatial correlation (the daily time series of each station are**
**considered in its calculation without any regional averaging).**

| Region | Number of stations | O₃ stations (µg/m3) | O₃ model (µg/m3) | NMB (%) | NRMSE (%) | r |
|---|---|---|---|---|---|---|
| China | 125 | 66 | 74 | 12 | 50 | 0.42 |
| CEC | 41 | 70 | 72 | 3 | 48 | 0.49 |
| BTH | 4 | 77 | 67 | -13 | 41 | 0.68 |
| YRD | 10 | 69 | 81 | 18 | 53 | 0.34 |
| PRD | 11 | 61 | 86 | 41 | 60 | 0.44 |

We use also IAGOS ozone profiles above Chinese and Korean airports for the period 2011-2017 to evaluate the model above 950 hPa (Fig. 2). The selected IAGOS data correspond to the lowermost troposphere (950-700 hPa), the lower (700-470 hPa) and upper (470-300 hPa) free troposphere and the UTLMS (<300 hPa) above the Chinese coast (east of 110°E and between 30 and 50°N) and South Korea. In order to assess more precisely the model abilities to reproduce the observed ozone behaviour, the IAGOS data are projected onto the model daily grid using the Interpol-IAGOS software (Cohen et al., 2020) and averaged every month. The subsequent product is called IAGOS-DM (Distributed onto the Model grid) hereafter. We derive monthly means from the INCA daily output by selecting the sampled grid cells. These monthly fields are called INCA-M (the M suffix referring to the IAGOS Mask). The two products IAGOS-DM and INCA-M are thus consistent in space and time, and can be compared together. It is important to note that the regional averages calculated here do not account for the tropopause altitude, in contrast to Cohen et al. (2020). Last, as in Cohen et al. (2018), the statistical representativeness of the observations is enhanced by filtering out the regional monthly means either with less than 300 data, or less than 7 days separating the first and the last measurements. A very good agreement is observed between the INCA model and the IAGOS observations with small biases ranging from 1.6% in the lower free troposphere to 12% in the lowermost troposphere. The INCA and IAGOS timeseries are well correlated with correlation coefficients equal or larger than 0.76. Looking in details the time series shows that the model tends to underestimate O₃ in the lowermost troposphere and to underestimate the largest O₃values in the lower and the upper free troposphere.



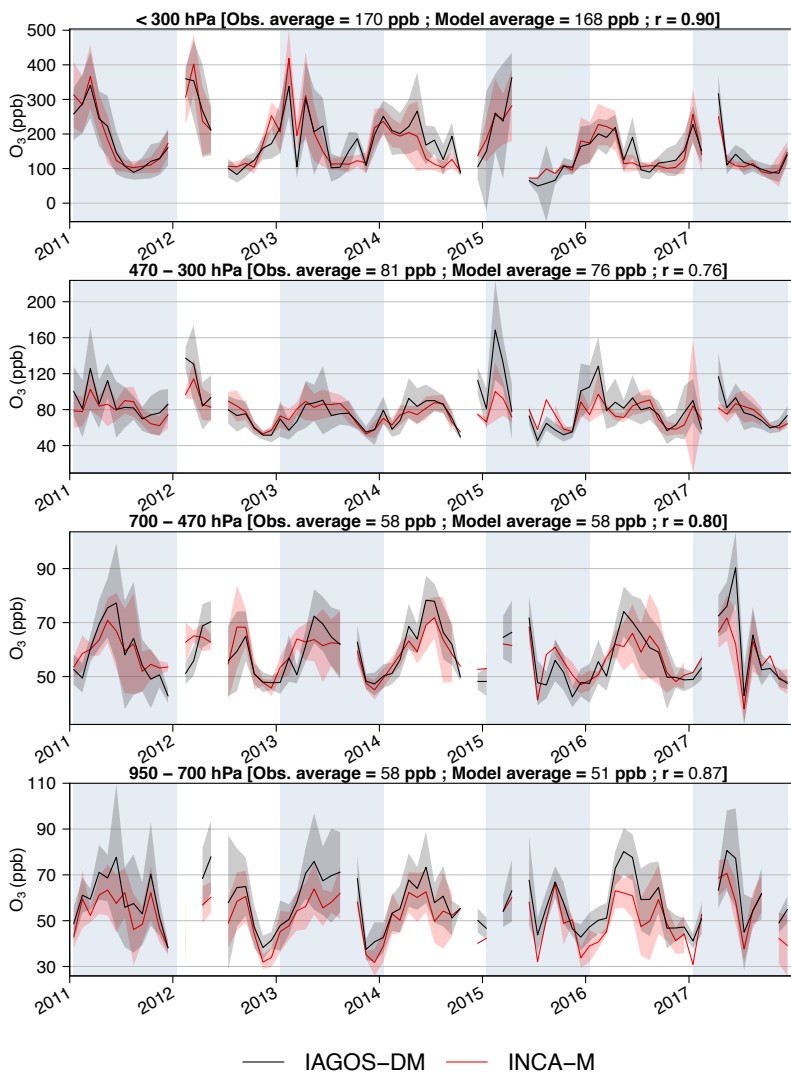

**Fig 2 Ozone monthly mean values derived in four altitude domains representing the LMT, LFT, UFT and UTLMS (from bottom to top) from the gridded IAGOS data set (solid black line) and the simulation output (solid red line) over Northeast Asia, with respect to the IAGOS mask. The uncertainties are defined by the regional average of the standard deviations calculated for each grid cell. For each altitude domain, the yearly O₃ concentration derived from the mean seasonal cycle is indicated above the corresponding graphic for both data sets, with the Pearson correlation coefficient comparing the two monthly time series.**



### 3.3 Comparison between IASI O₃ product and INCA simulation in North East Asia

We compare IASI and INCA O$_3$ partial columns over the East Asia domain (100-145°E, 20-48°N) averaged over the 2008-2017 period, the model being smoothed (Fig. A2) or not (Fig. 3) by IASI averaging kernels. Spatial distribution and spatial gradients of ozone are in good agreement for the 4 partial columns. On average, the differences are smaller than 5% for the 0-3km and 3-6km columns (1.5% and 2.7% respectively without AK smoothing). The difference is larger for the upper columns - 6-9km and 9-12km columns – with a mean negative difference of about 15 %, INCA being larger than IASI (-14.9% and -16.4% respectively). The agreement is improved when the model is smoothed by the IASI averaging kernels, the mean difference is reduced to -7% for both 6-9km and 9-12km columns. For the 0-3km partial column, it is worth noting that the IASI retrieval is not highly sensitive to these altitudes and that the a priori contribution is larger (Dufour et al., 2012). However, the agreement between IASI and INCA remains largely reasonable accounting for the observation and model uncertainties. IASI is systematically smaller than INCA over China ranging from -5% to -25% (Fig. 3a) and the agreement improves within +/-5% when applying the AK to the model (Fig. A2a). A difference smaller than 10% is observed over Korea, Japan, and the surrounded seas. Larger differences are seen for tropical maritime regions largely reduced when AK are applied. This reflects the reduced sensitivity and larger a priori impact of IASI retrievals in the lowest layers. For the 3-6km partial columns, where the IASI retrievals are the most sensitive, a very good agreement between IASI and INCA, within +/-10%, is observed for a large part of the domain (Figs 3b and A2b). It is the partial columns for which the agreement is the best. For the upper columns (6-9km and 9-12km), IASI is almost systematically smaller than INCA over the domain (Figs 3c-d and A2c-d). IASI is always smaller than INCA over the most part of China whatever the partial columns considered. IASI is mainly larger than INCA in the lower troposphere and smaller in the upper troposphere elsewhere. In the desertic northwestern part of the domain, even if the emissivity is included in the IASI retrievals, the quality of the retrievals can be affected and confidence in the data reduced. This region should then not be considered here. The retrieval in the tropical-type airmasses have been shown to reinforce the natural S-shape of the ozone profiles, leading to some overestimations of ozone in the lower troposphere and an underestimation in upper troposphere (Dufour et al., 2012). This likely explains the positive and negative differences with the model in the southeastern part of the domain (Fig. 3). This translates even stronger to the model when AKs are applied: the model is then smaller than IASI in the upper troposphere (Fig. A2). Globally, the differences between IASI and INCA are the smallest over the Central East China (CEC). Figure 4 shows the IASI and INCA monthly timeseries of the different O$_3$ partial columns between 2008 and 2017 for this region. The correlation between the IASI and INCA timeseries is good: larger than 0.8 except for the 6-9km column (0.75). The high correlation is partly driven by the seasonal cycle, but the correlation remains quite high for deseasonalized (anomalies) series – 0.65, 0.63, and 0.68 for 0-3km, 3-6km and 9-12km columns respectively – except for the 6-9km column (0.44). Biases ranging from 8% to 14%, INCA being larger, are observed between IASI and INCA for the 0-3km, 6-9km and 9-12km columns respectively. The highest values are larger with INCA for the 0-3km and 9-12km columns and the lowest values larger for the 6-9km columns (Fig. 4). A smaller bias (-3.4% on average) better balanced between small and large values is observed for the 3-6km column (Fig. 4b). The seasonal cycle observed with IASI is





reasonably reproduced by the model for the different partial columns with a better agreement in the 3-6km and 9-12km columns. However, the summer drops observed with IASI in the lower troposphere (0-3km and 3-6km) is not systematically reproduced by the model and the summer maximum is shifted for the 6-9km column.

In the following, after presenting the trend analysis globally over the Asian domain for the different partial columns, the
discussion will focus more on the 3-6km partial column where IASI and INCA agree well. The CEC region will also be privileged in the discussion as the model and observation operate better and they are in rather good agreement in this region. Some other highly populated and polluted regions such as the Sichuan Basin (SCB) and the Pearl River Delta (PRD) will be also discussed keeping in mind the largest differences between model and observations. As we show here, comparison between IASI and INCA are satisfying with and without applying the AK to the model. For the trend analysis, we will consider the
model without AK applied to avoid introducing retrieval a priori information in the model and have a model fully independent of the observations. This will allow us to exploit the sensitivity tests conducted with the model to determine the processes that drive the trends.

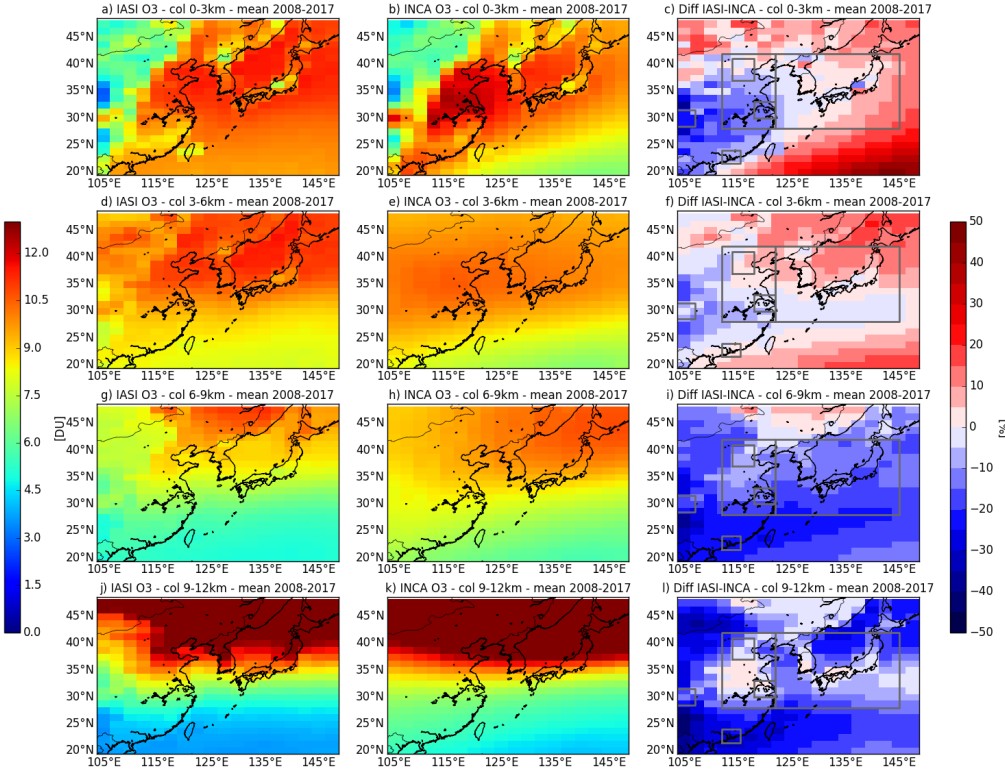

**Figure 3: Mean O$_3$ partial columns for 2008-2017 observed with IASI (left panels), simulated by INCA (middle panels), and their**
**differences (right panels). For different partial columns are considered: the lowermost tropospheric columns from the surface to 3 km altitude (named 0-3km), the lower free tropospheric columns from 3 to 6 km altitude (named 3-6km), the upper free tropospheric column from 6 to 9 km (named 6-9km), and the upper tropospheric column from 9 to 12 km (named 9-12km).**



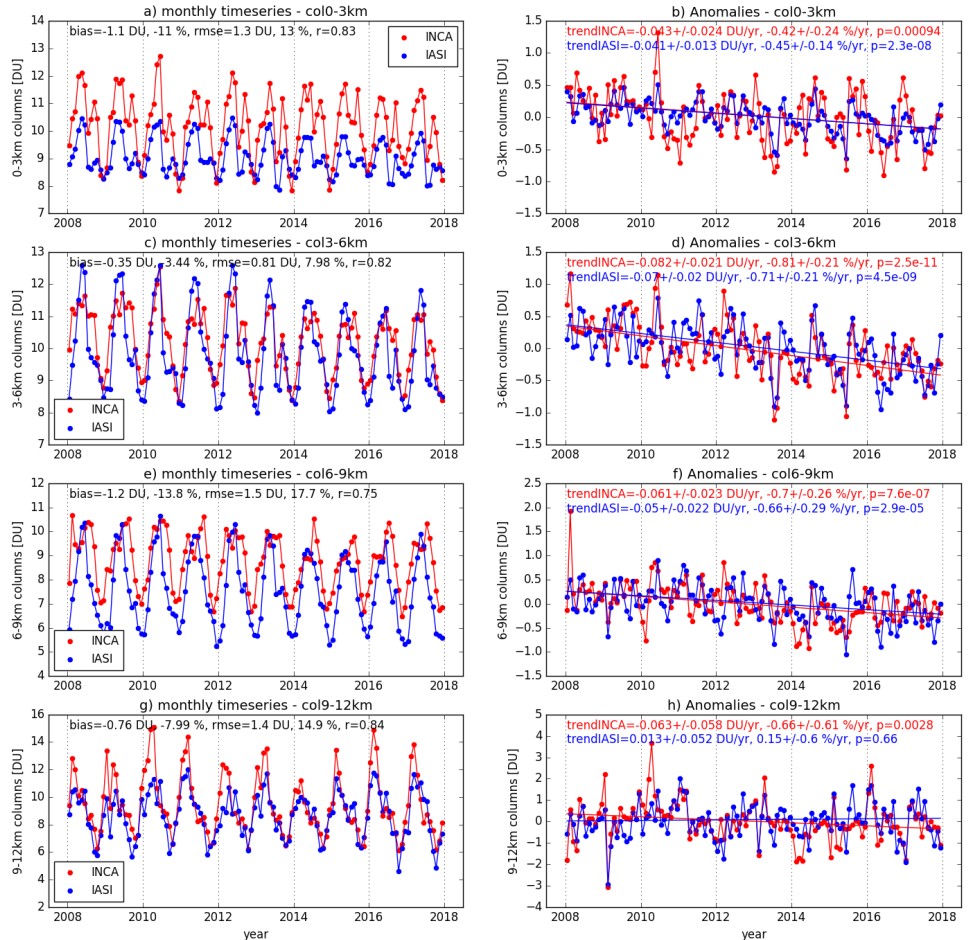

**Figure 4: IASI and INCA monthly timeseries (left) and anomalies (right) for the 0-3km, 3-6km, 6-9km, and 9-12km O₃ partial columns over the CEC region for 2008-2017. Biases, RMSE and pearson correlation coefficient between the IASI and INCA monthly timeseries are provided as well as the trends, uncertainties, and associated p-values calculated from the anomalies timeseries.**

## 4 O₃ trends: satellite and model comparison

To derive the trends, we first calculate the monthly timeseries either at the INCA resolution – gridding IASI at this resolution – or averaging the model or observation partial columns over the regions reported in Fig. 1. The monthly mean ozone values are used to calculate a mean 2008-2017 seasonal cycle. This cycle is then used to deseasonalize the monthly mean timeseries by calculating the anomalies. The linear trend is then calculated based on the monthly anomalies. It is provided either in DU/yr or in %/yr. The trends uncertainties correspond to the 95% confidence interval, the p-values are also calculated and reported when possible. An example is given in Fig. 4 for the CEC with monthly timeseries on the left and anomalies on the right.





Figure 5 presents 2008-2017 $O_3$ trends derived for different partial columns over the Asian domain using IASI and INCA. Trends derived from IASI are negative with p-value < 0.05 for most of the domain and the different partial columns, except in the upper troposphere (9-12km column). They range between -0.2 %/yr and -0.6 %/yr for the 0-3km column, and between -0.4 %/yr and -1 %/yr for the 3-6km and 6-9km columns. The trends derived from the model are rather uniform over the domain

for the 0-3km and 3-6km columns, being smaller than -1%/yr, except in the CEC region where the trends tend to zero in the lowermost troposphere (0-3km). It is worth noting that the model shows positive trends at the surface level in this region (not shown) in agreement with surface measurement studies (e.g. Li et al., 2020). A residual positive trend is observed up to 1 km altitude in the model and becomes negative higher (not shown). The trends in the mid-upper troposphere (6-9km and 9-12 km columns) are mainly negative (p<0.05) north to 30°N latitude and can be more variable in the subtropics (Fig. 5). To evaluate

the impact of the IASI sampling (representative of clear-sky conditions), we calculate the model monthly mean including the model grid cells on the days when IASI observations are available in these cells. The trends derived from the model resampled to match IASI observations are reported on Fig. 5. The resampling changes only slightly the trends derived from the model. In the following, we consider the model without matching the IASI sampling.

Table 3 summarizes $O_3$ trends derived from IASI and INCA for different partial columns and for the different regions reported

in Fig. 1. We choose the most populated Chinese areas where significant pollutant reductions have occurred since 2013 (Zheng et al., 2018), such as CEC – including BTH and YRD – and PRD and SCB. We also consider the KJ region as a region influenced by the pollution export from China. We bold the trend values in the table when both IASI and INCA have trends with p<0.05 and when the trends agree within 40% between the model and the observations. The trend values corresponding to p<0.05 and a poorer agreement are in italic. The CEC region shows the best agreement between the trends derived from

IASI and INCA for all the columns except the upper tropospheric columns (9-12km). The anomalies and calculated linear trends are shown in detail on Fig. 4 (right panels). For this region, where both the observations and the model are the most reliable, trends are in very good agreement (<15%) for the 0-3km, 3-6km and 6-9km columns. Trends derived from IASI for the UTLMS columns (9-12km) are very small with large p-values for all the regions (Table 3). It is then difficult to compare and conclude for the upper tropospheric columns - the trends calculated from the model are mainly negative with p<0.05. For

the PRD and the SCB, the model and the observations are less reliable for different reasons explained in section 3. This leads to a poor agreement of the derived trends and a lack of reliability of the trends for these two regions (large uncertainties on the trends values, Table 3). For the BTH and YRD, included in the CEC, and for the KJ, the trends calculated from the observations and the model are in good agreement for the 3-6km and 6-9km columns with p<0.01.



**Table 3: Calculated trends in DU/yr from IASI observations and INCA simulations for the different regions of Fig. 1 and the 0-3km, 3-6km, 6-9km, and 9-12km partial columns. The associated p-value is indicated for each trend. The trend values are in bold when both IASI and INCA trends have associated p < 0.05 and are within 40% agreement. The trend values are in italic when both IASI and INCA trends have associated p < 0.05 but with differences larger than 40%.**

| | 0-3km | | 3-6km | | 6-9km | | 9-12km | |
|---|---|---|---|---|---|---|---|---|
| | IASI | INCA | IASI | INCA | IASI | INCA | IASI | INCA |
| CEC | **-0.04 ± 0.01** (p<0.01) | **-0.04 ± 0.02** (p<0.01) | **-0.07 ± 0.02** (p<0.01) | **-0.08 ± 0.02** (p<0.01) | **-0.05 ± 0.02** (p<0.01) | **-0.06 ± 0.02** (p<0.01) | 0.01 ± 0.05 (p=0.66) | -0.06 ± 0.06 (p<0.01) |
| BTH | -0.05 ± 0.01 (p<0.01) | -0.02 ± 0.01 (p=0.16) | **-0.09 ± 0.02** (p<0.01) | **-0.09 ± 0.01** (p<0.01) | **-0.06 ± 0.03** (p<0.01) | **-0.09 ± 0.04** (p<0.01) | 0.02 ± 0.07 (p=0.61) | -0.11 ± 0.10 (p<0.01) |
| YRD | *-0.03 ± 0.02* *(p=0.04)* | *-0.06 ± 0.04* *(p<0.01)* | **-0.05 ± 0.03** (p<0.01) | **-0.08 ± 0.03** (p<0.01) | **-0.05 ± 0.03** (p<0.01) | **-0.05 ± 0.03** (p<0.01) | -0.009 ± 0.06 (p=0.54) | -0.02 ± 0.05 (p=0.05) |
| KJ | *-0.03 ± 0.01* *(p<0.01)* | *-0.09 ± 0.03* *(p<0.01)* | **-0.06 ± 0.02** (p<0.01) | **-0.08 ± 0.02** (p<0.01) | **-0.05 ± 0.02** (p<0.01) | **-0.06 ± 0.03** (p<0.01) | -0.01 ± 0.05 (p=0.49) | -0.05 ± 0.06 (p=0.03) |
| PRD | -0.04 ± 0.02 (p<0.01) | -0.05 ± 0.06 (p=0.09) | -0.06 ± 0.02 (p<0.01) | -0.03 ± 0.04 (p=0.22) | -0.04 ± 0.02 (p<0.01) | -0.03 ± 0.04 (p=0.15) | 0.009 ± 0.03 (p=0.67) | -0.04 ± 0.03 (p=0.02) |
| SCB | -0.007 ± 0.01 (p=0.07) | -0.02 ± 0.006 (p<0.01) | *-0.02 ± 0.02* *(p<0.01)* | *-0.08 ± 0.03* *(p<0.01)* | -0.02 ± 0.02 (p=0.05) | -0.02 ± 0.03 (p=0.24) | 0 ± 0.04 (p=0.89) | 0.02 ± 0.03 (p=0.23) |





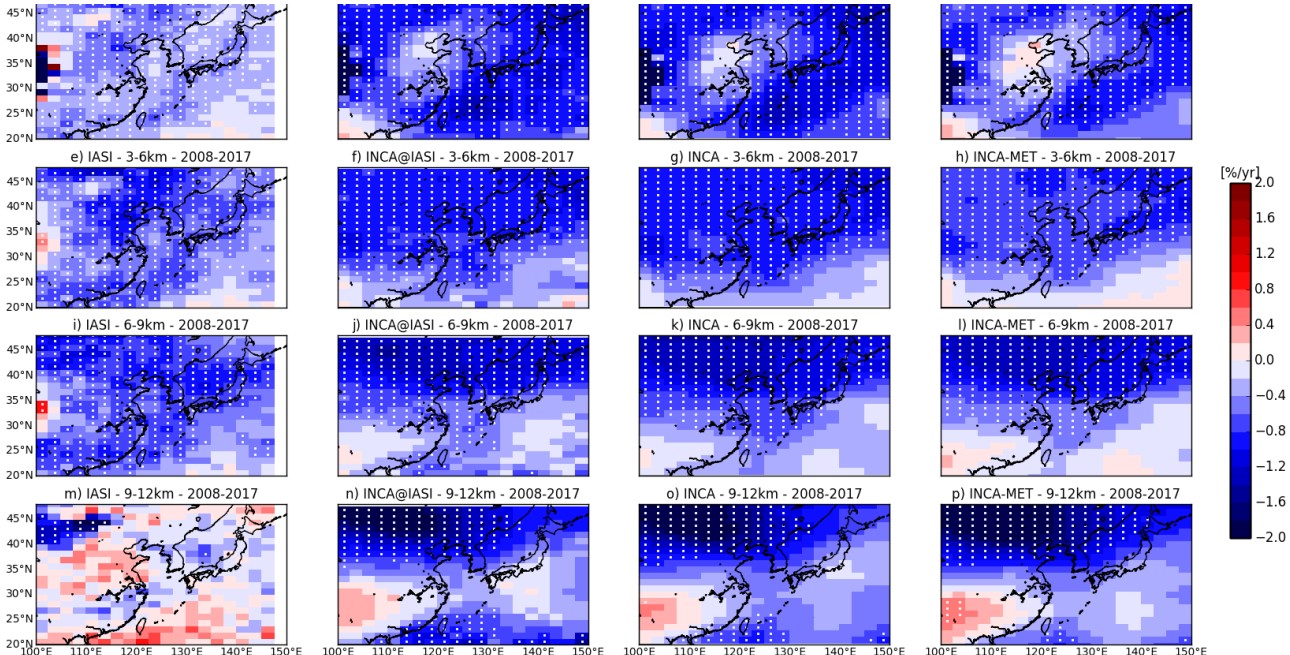

**Figure 5: Trends calculated in %/yr at the 2.5°x1.25° resolution for the 0-3km, 3-6km, 6-9km, and 9-12km O₃ partial columns. (a,e,i,m) trends derived from IASI, (b,f,j,n) trends derived from INCA sampled over the IASI pixels, (c,g,k,o) trends derived from INCA with its native daily resolution, (d,h,l,p) trends derived from the INCA reference simulation minus the MET simulation (see text and Table 4 for details). White crosses are displayed when p-values are smaller than 0.05.**

## 5 Discussion

### 5.1 Sensitivity tests to evaluate the processes contributing to the trends

In this work focused on China and 2008-2017, both IASI observations and INCA simulations show negative O₃ trends of similar magnitude in the lower (3-6km) and upper (6-9km) free troposphere in large parts of China and its downwind region. Dufour et al. (2018) suggest that negative O₃ trends derived from IASI in the lower troposphere over the North China Plain for a slightly shorter time period can be explained for almost half by NO$_x$ emissions reduction in China since 2013. They argue the negative impact on the trend of these reductions compared to the positive one at the surface is due to changes in the chemical regime with the altitude. To go further on this and quantify the processes contributing to the O₃ trends, we performed, as described in Section 2, several sensitivity simulations with the INCA model. The objective is to remove one-by-one the interannual variability and trend induced by the different processes (emissions, meteorology). The different sensivity simulations are summarized in Table 4. First, anthropogenic emissions in China are kept constant to their 2007 values during





the 2008-2017 period. Then, in addition, the global anthropogenic emissions for the rest of the world are kept constant to their 2007 values, and the biomass burning emissions, and then methane concentrations. Finally, winds used for nuding and sea-surface temperatures are maintained at their 2007 values. Since the INCA model is a GCM this method reduces considerably the interannual variability associated with meteorology. However, the meteorology is not identical from one year to another in

5   the GCM. This leads to a residual trend shown in Fig. B1 (Appendix B) for the different partial columns. This residual trend is mainly negative and small. To remove this residual trend from the model results, we subtract the MET simulation to the other simulations day by day and grid cell by grid cell. The trends derived from the Ref-MET simulation are slightly increased compared to the trends derived from the Ref simulation as the residual negative trends have been removed but they remain fully consistent (Fig. 5), especially when $p<0.05$. Then, we calculate the contribution of each process as described in Table 4.

**Table 4: List and conditions of the different sensitivity tests realized with the INCA model. Note that for all the simulations the biogenic emissions are constant over the period.**

| Simulations | Sensitivity test conditions | Contribution calculation |
|---|---|---|
| Reference (Ref) | Incl. Chinese changing emissions as described in section 2.6 | |
| Chinese emissions (China) | Ref simulation but with constant Chinese anthropogenic emissions set to 2007 | Contribution China = (ref –China)/ref |
| Global emissions (Global) | China simulation but with constant global anthropogenic emissions set to 2007 | Contribution Global = (China –Global) / ref |
| Biomass burning (BBg) | Global simulation but with constant biomass burning emissions set to 2007 | Contribution BBg = (Global –BBg)/ref |
| Methane ($CH_4$) | BBg simulation but with constant $CH_4$ concentrations set to 2007 | Contribution $CH_4$ = (BBg – $CH_4$) / ref |
| Meteorological (MET) | $CH_4$ simulation but with winds and SST set to 2007 | Contribution MET = $CH_4$ / ref |

The contribution of the different processes is shown in Fig. 6 for the 3-6km and 6-9km columns. We focus on these two

15   columns as the trends are in good agreement between IASI and INCA. The main contributions to the trends are the local Chinese emissions and the meteorology, with contributions larger than 20%. The other tested variables (global emissions, biomass burning emissions and methane) contribute to the trends within 20%, with a negative contribution of methane for most of the domain. This means that the increase in methane concentrations and then the associated ozone production counteracts the ozone reduction due to the other processes (emissions and meteorology). The different contributions for the

20   regions where the model and the observations are the most reliable are detailed in Table 5. Chinese emissions contribute to 60% in the main source region, the CEC, with variations inside the regions for the 3-6km column: the Chinese emissions contribute to 40% in BTH and more than 70% in YRD. The Chinese contribution to the trends in the export region (KJ) remains high with 47% contribution. The meteorological contribution ranges from 34 to 38%. Methane and biomass burning emissions contributions are rather stable over the different regions around -15% and +14% respectively, biomass burning contribution



being slightly higher in the export region (19% for KJ). Surprisingly, the global emissions contribute the most to the trends in the highly polluted region of the BTH (22%). For the 6-9km column, the meteorological contribution to the trends increases (about 50% or larger) as the Chinese emissions contribution decreases. The biomass burning contribution is globally larger, especially in the export region where it reaches 30%. The global anthropogenic emissions contribution remains small in absolute value, except in the YRD region where it becomes negative and reaches -20%. The prevailing contribution of Chinese emissions changes in the negative O3 trends in the lower and upper free troposphere seems to confirm the previous outcomes of Dufour et al. (2018).

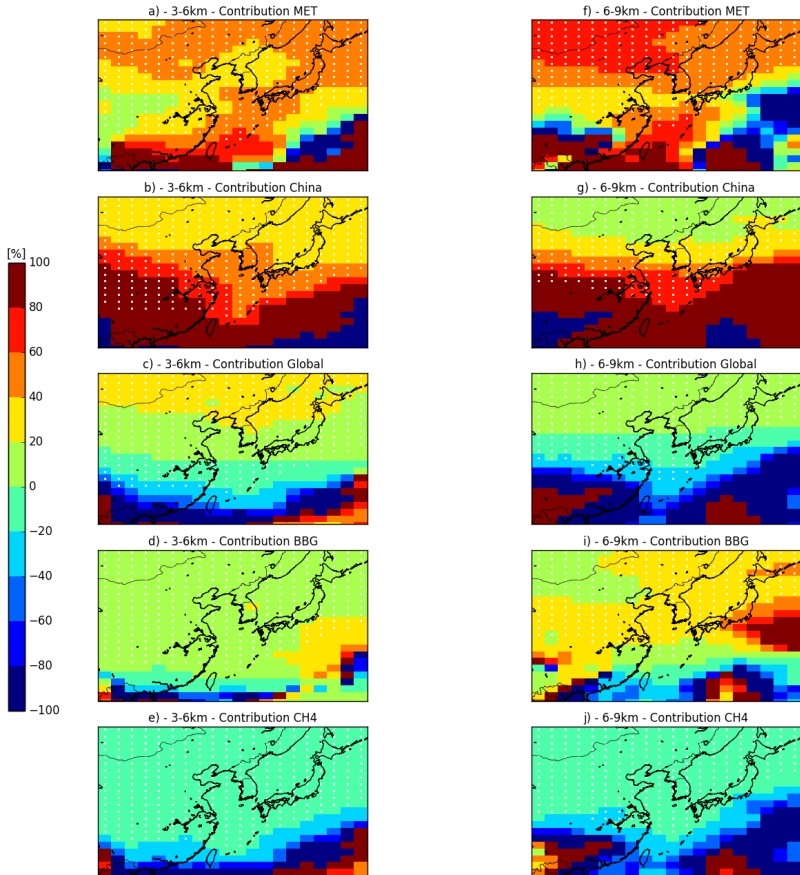

**Figure 6: Contributions of the meteorological variability, the Chinese anthropogenic emissions, the global anthropogenic emissions, the global biomass burning emissions and the CH₄ to the trends calculated for the 3-6km and 6-9km partial columns (see Table 4 for details on the sensitivity tests).**



**Table 5:  Contributions (in %) of the meteorological variability, the Chinese anthropogenic emissions, the global anthropogenic emissions, the global biomass burning emissions and the CH₄ to the trends calculated for the 3-6km and 6-9km partial columns for each individual region in Fig. 1.**

|  | 3-6km partial column | | | | | 6-9km partial column | | | | |
| --- | --- | --- | --- | --- | --- | --- | --- | --- | --- | --- |
|  | MET | CH$_4$ | BBg | Global | China | MET | CH$_4$ | BBg | Global | China |
| CEC | 34.3 | -15.2 | 13.4 | 7.4 | 60.1 | 50.0 | -14.9 | 18.1 | -5.4 | 52.2 |
| BTH | 37.8 | -14.5 | 14.7 | 21.9 | 40.1 | 55.4 | -11.4 | 17.0 | 4.1 | 34.8 |
| YRD | 34.2 | -15.9 | 13.1 | -2.5 | 71.0 | 42.0 | -17.4 | 14.7 | -18.7 | 79.5 |
| KJ | 37.9 | -15.2 | 18.6 | 11.8 | 46.9 | 46.1 | -15.6 | 30.2 | -3.4 | 42.7 |

## 5.2 Limitation of the study

The results presented in our study are not fully in line with the recent Tropospheric Ozone Assessment Report (Gaudel et al., 2018) and related works (Cooper et al., 2020; Gaudel et al., 2020), which states a general increase in tropospheric ozone during the last decades. If negative trends are observed at the surface in developed countries for example in summer, positive trends for the free troposphere are reported using mainly IAGOS as a reference (Cooper et al., 2020). Even if our study seems to show consistent trends derived by IASI and the model in the free troposphere, we stress, in this subsection, vigilant points for the interpretation of the results.

*Length of the period*

In this study, we derived trends over a limited 10-year period. Calculating short-term trends leads to an increased sensitivity to the inter and intra-annual variations, the length of the period and to the starting and ending point of the time series. Due to the availability of the satellite measurements and the simulated period with the model, it was not possible to extend the time period further. We tested the impact of the starting and ending point of the time series by removing one and two years at the beginning and at the end of the period. The trends derived from IASI and INCA for the 3-6km column in the CEC for the different periods are summarized in Table B1 (Appendix B). They remain consistent with the trend derived for 2008-2017, respectively for IASI and INCA, and within its confidence interval. These results seem to comfort the consistency between the modelled and observed trends and their robustness. However, it is worth noting that the calculated trends seem more sensitive to the end of the period, corresponding to a strong El Nino period, than to the beginning of the period. Removing the last two years of the period leads to a decrease of the IASI trend and to an increase of the INCA trend. This apparent inconsistency, which should be evaluated when longer simulations with consistent emissions and longer observation time series will be available, stress the difficulty of working with short-term trends and the caution to take to not overinterpret the results.



*Discrepancies between different satellite sounders and products*

The TOAR points toward a major discrepancy between the different satellite ozone products available for the report: the OMI UV sounder showing mainly positive trends over 2008-2016 and the IASI IR sounder showing mainly negative trends over the same period (Fig B2 – Appendix B). Over North East Asia, the discrepancy in the sign of the trend calculated from the different satellite products is more contrasted. The OMI/MLS and OMI-RAL products still show positive trends as all over the globe. The IASI-SOFRID product shows positive trends all over China, and the IASI-FORLI product positive trends only in the southeastern part of China. The IASI $O_3$ product used in this study was not included in the comparison as it is not a global product. It is worth noting that for the TOAR, the tropospheric columns derived from the different satellite products were not based on the same tropopause height definition and each product was considered with its native sampling. This might contribute partly to the differences between the trends, in addition to the fundamental differences in the measurement techniques (UV and IR) and the retrieval algorithms used. Possible drifts over the time have not been systematically studied in the TOAR. Some individual studies exist but once again they do not allow one to conclude. Indeed, Boynard et al. (2018) noticed a significant negative drift in the Northern Hemisphere in the IASI-FORLI product, which is not detected in the most recent IASI-SOFRID product (Barret et al., 2020). The OMI/MLS product shows a small positive drift when compared to ozonesondes but not significant when based upon a difference t-test (Ziemke et al., 2019). For this study, we compare our IASI $O_3$ product with the OMI/MLS one. To conduct a proper comparison, we used the same definition of the tropopause height to calculate the tropospheric columns. As the OMI/MLS product provides directly tropospheric columns without ozone profiles, we selected the tropopause height used for OMI/MLS, derived from the NCEP re-analyses, as the reference tropopause height. We calculated the tropospheric columns from the IASI $O_3$ profiles retrieved up to the defined tropopause height. We calculated the monthly time series at the resolution of 1°x1°. Only the days for which IASI and OMI/MLS are both available in the considered grid cell are used to calculate the monthly means and anomalies for the given grid cell. The derived trends are shown in Fig. 7. OMI/MLS shows large positive trends all over the domain except in the southern part of the BTH region. On its side, IASI shows trends close to zero with positive trends over central China, the East China Sea and over the Pacific in the southeastern part of the domain, and negative trends over North China and Korea. The TOC trends derived from IASI show completely different spatial patterns from the trends derived in the free troposphere (Fig. 5) and seem to reflect more the trends of UTLS column (9-12 km). Work is still needed to understand the differences in the trends derived from different satellite instruments. Especially, one important question is to identify from which part of the troposphere the TOC is the most representative in the different products and how the vertical sensitivity of the different instruments and retrieval algorithms influence the calculated columns and trends. Answering this question is one of the objectives of the satellite working group of the TOAR phase II, which started in 2021.



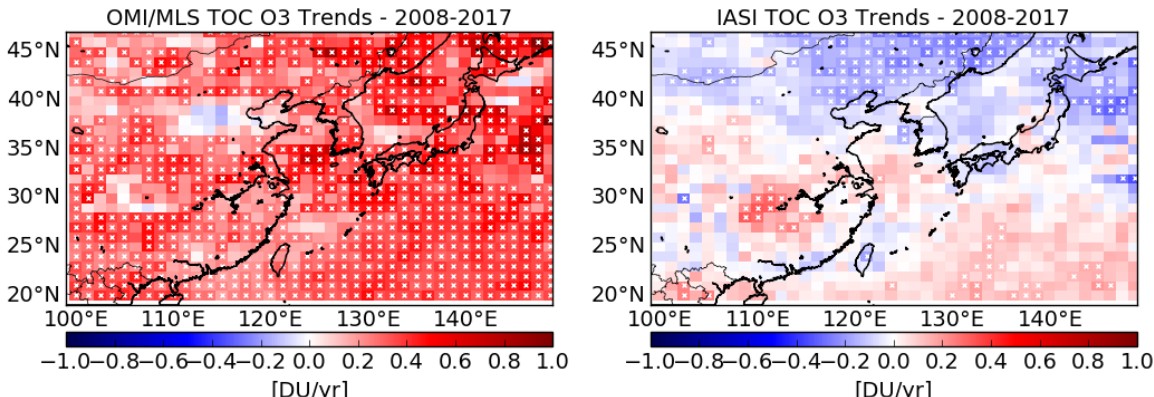

**Figure 7: O₃ trends derived for the Tropospheric O₃ Column (TOC) from OMI/MLS and IASI for 2008-2017 at 1°x1° resolution. The TOC is calculated using the same tropopause height for OMI/MLS and IASI. White crosses indicate associated p-values smaller than 0.05.**

*Impact of the sampling: comparison with IAGOS*

The IAGOS observations are considered as a reference for free tropospheric ozone trends in the TOAR framework. Then, we try to evaluate the IASI and INCA trends using IAGOS in addition to its use for validation reported in section 3. As already mentioned for the validation of IASI using IAGOS, the comparison is somehow difficult due to the limited top altitude of
IAGOS profiles to properly apply the AK to the profiles. We limit our comparison to the 3-6km columns and explore the impact of the sampling on trend calculations. We use the same coincidence criteria than the one used in section 3 to select pairs of IAGOS and IASI profiles. Based on the selected profiles, we consider the daily simulated profiles of INCA for the same day and the grids cells of the model corresponding to the latitudes and longitudes of the profiles. Partial columns are then calculated for the subset of observed and modeled profiles and the trends are derived from the monthly anomalies. As
mentioned in section 3, the number of IAGOS profiles in the Chinese region is not very high (about one thousand) and reduces to 315 profiles in coincidence with IASI with 213 profiles covering the pressure range from 1000 hPa to 500 hPa. This number even falls to 26 profiles for profiles within 1000 hPa and 250 hPa. The trends that can be calculated from this set of profiles are then not robust enough to conclude. Then, we consider the European region for which more profiles are available. 9185 IAGOS profiles are initially available for 2008-2017 with almost no measurements in 2010. Looking for the subset of profiles
in coincidence with IASI strongly reduces the number of available profiles: 3276 are selected. We consider IAGOS profiles covering the 1000-250hPa range for a proper comparison with IASI. This allows one to reduce the proportion of a priori information potentially introduce in the IAGOS profiles when they are completed up to 60 km and smoothed by the AKs (see section 3 for details). This leads to reduce the subset of profiles to 1103 profiles. Table 6 provides the trends derived from this subset of profiles over Europe for IASI, INCA and IAGOS. IAGOS trends are calculated from both raw and smoothed profiles.
We also provide the IAGOS trends calculated from the initial set of IAGOS profiles to evaluate the sampling impact. We





calculate the trends for 2008-2017 and 2011-2017 as 2010 was not sampled by IAGOS and that 2008-2009 were associated to a negative anomaly (Cooper et al., 2020) which might perturb the short-term trend calculation. It is interesting to note that the difference in the sampling between the initial set of IAGOS profiles and the set in coincidence with IASI changes the calculated trends from a positive trend (0.05 DU/yr, p<0.01) to a trend close to zero for both the raw and smoothed IAGOS columns

5    (0.002 DU/yr, p=0.94 and 0.006 DU/yr, p=0.81) for 2008-2017. For comparison, the IASI and INCA trends are negative (-0.09 DU/yr, p=0.01 and -0.065 DU/yr, p=0.81) for the same period. For 2011-2017, no trend is reported from the initial set of IAGOS columns (0.001 DU/yr, p=0.95). The trends turn to be negative when considering the subset of IAGOS columns in coincidence with IASI (-0.05 DU/yr, p=0.26 for raw columns and -0.07 DU/yr, p=0.05 for smoothed columns). The IASI and INCA trends remain negative for this time period, -0.14 DU/yr (p<0.01) and -0.06 DU/yr (p=0.04) respectively. The IASI

10   negative trend is more than twice larger in absolute value compared the ones derived from IAGOS and INCA. These results do not allow one to clearly conclude wether the negative trends derived from the model and IASI are realistic or not. They mainly show the strong sampling issue for trend calculation and stress the need to compare different datasets using, as far as possible, similar sampling to evaluate the derived trends. These results highlight again the difficulty to draw firm conclusions on short-term trends derived from different datasets in addition to sampling differences.

Table 6: **Trends calculated for the 3-6km O$_3$ columns in Europe using a subset of coincidence profiles of IASI, IAGOS and INCA (see the text for the details on the coincidence criteria) and using the initial set of IAGOS profiles. The IAGOS trends for the coincident set of profiles are calculated from the raw and the smoothed profiles. The trends are calculated for two periods 2008-2017 and 2011-2017**

|  | 2008-2017 | 2011-2017 |
|---|---|---|
| IASI coincident | -0.09 ± 0.07 (p=0.01) | -0.14 ± 0.10 (p<0.01) |
| INCA coincident | -0.065 ± 0.04 (p<0.01) | -0.06 ± 0.06 (p=0.04) |
| IAGOS initial sampling | 0.05 ± 0.03 (p<0.01) | 0.001 ± 0.04 (p=0.95) |
| IAGOS coincident and raw | 0.002 ± 0.05 (p=0.94) | -0.05 ± 0.08 (p=0.26) |
| IAGOS coincident and smoothed | 0.006 ± 0.05 (p=0.81) | -0.07 ± 0.07 (p=0.05) |

## 6 Conclusions

In this study, the tropospheric ozone trends in China and export regions are investigated for 2008-2017 using the IASI O$_3$ observations and the LMDZ-OR-INCA model simulation, including the most recent Chinese emission inventory (Zheng et al., 2018). We focus mainly on the lower (3-6km) and upper (6-9km) free troposphere where the IASI observations and the model

25   simulations are in good agreement, especially in the Central East China (CEC) region. These vertical layers correspond also to the atmospheric regions where the IASI observations are the most sensitive. In addition, the evaluation of the model based on surface measurements and IAGOS observations shows good performances of the model from the surface to the UTLMS in



the CEC. The $O_3$ trends calculated from the IASI observations and the INCA model are in very good agreement in the CEC region and subregions such as BTH (Beijing-Tianjin-Hebei) and YRD (Yangtze River Delta) included in the CEC. The observed and modeled trends in the CEC region are -0.07 ± 0.02 DU/yr and -0.08 ± 0.02 DU/yr respectively for the lower free troposphere, and -0.05 ± 0.02 DU/yr and -0.06 ± 0.02 DU/yr respectively for the upper free troposphere. A good agreement is

also observed in the region including Korea and Japan and corresponding to the region of pollution export from China. Based on sensitivity studies conducted with the INCA model, we quantify the contribution of the Chinese anthropogenic emissions, the global anthropogenic emissions, the global biomass burning emissions, methane, and meteorology to the ozone trends. In the CEC region, 60% of the negative trend derived from the model in the lower free troposphere can be attributed to the Chinese anthropogenic emissions and 52% in the upper free troposphere. The second contribution to explain the negative trend

is the meteorological variability (34% and 50% respectively). The background ozone produced from methane globally counteracts the decrease in ozone with a contribution of about 15 % to the trends in the lower and upper free troposphere. The global anthropogenic emissions changes account for less than 10% in the ozone trends and biomass burning emissions changes between 10 and 20 %. These results suggest that the reduction of $NO_x$ anthropogenic emissions that occurs since 2013 in China leads to a decrease in ozone in the Chinese free troposphere, contrary to the increase in ozone at the surface. However, too few

independent measurements such as IAGOS or ozonesondes are available in the region during 2008-2017 to fully validate the decreasing trends calculated by both the IASI observations and the model. A comparison done in Europe where more independent IAGOS measurements are available show that trend calculation can be strongly affected by the sampling of the considered datasets and the time period considered when analyzing short-term trends. Particular caution should be taken to not overinterpret short-term trends and when comparing trends derived from different datasets with different sampling. In addition,

comparisons between the trends calculated from the OMI/MLS $O_3$ tropospheric columns and the IASI ones, calculated using the same tropopause height and sampling, show large discrepancies as already stated by the TOAR (Gaudel et al., 2018), and point toward a need to better understand how the differences in vertical sensitivity of the satellite observations impact the observed tropospheric columns and the derived trends.

**Data and code availability**

The IAGOS data set is available at https://doi.org/10.25326/20, and more precisely, the time series data are available at https://doi.org/10.25326/06. The distribution of the IAGOS data onto the model grid is based on an updated version of the Interpol-IAGOS software, which can be found at https://doi.org/10.25326/81.

The LMDZ, INCA and ORCHIDEE models are released under the terms of the CeCILL license. The mode codes, input data, and outputs are archived in the CEA (Commissariat à l'énergie atomique et aux énergies alternatives) high- performance

computing centre TGCC and are available upon request.



The IASI observations (level 1C) are available from the AERIS data infrastructure (www.aeris-data.fr). The full archive of IASI ozone product retrieved from the level 1C data is available, upon requested to Gaëlle Dufour (gaelle.dufour@lisa.ipsl.fr), for the Asian domain considered here between 2008 and 2017.

In addition, the monthly gridded partial columns derived from IASI and INCA, used to calculate the trends in the project, will
be made available through a DOI (DOI attribution under request).

**Author contribution**

GD managed the study from its conception, the analysis of data, the preparation of the manuscript and the funding acquisition. DH and YZ performed the model simulations. ME performed the IASI ozone retrieval and managed the resulting level-2 product. YC, AG, and VT provided IAGOS observations and helped with their use and analyses in the study. BB was in charge
of ground surface data processing and cleaning. GS, ML, and AB managed the model evaluation with the surface measurements. JZ provided the OMI/MLS satellite data. All the authors participated in reviewing and editing the manuscript.

**Competing interests**

The authors declare that they have no conflict of interest.

**Acknowledgements**

The authors are grateful for the essential support of the Agence Nationale de la Recherche (ANR) through the PolEASIA project (ANR-15-CE04-0005). The IASI mission is a joint mission of EUMETSAT and the Centre National d'Etudes Spatiales (CNES, France). This study was financially supported by the French Space Agency – CNES (project "IASI/TOSCA"). The authors acknowledge the AERIS data infrastructure (https://www.aeris-data.fr) for providing access to the IASI Level 1C data, distributed in near-real-time by Eumetsat through the EumetCast system distribution. The authors acknowledge dataset
producers and providers used in this study: the ozonesonde data used in this study were mainly provided by the World Ozone and Ultraviolet Data Centre (WOUDC) and are publicly available (see http://www.woudc.org). We acknowledge the Institut für Meteorologie und Klimaforschung (IMK), Karlsruhe, Germany, for a licence to use the KOPRA radiative transfer model. A part of this work was performed using HPC resources from GENCI-TGCC (Grant A0050106877 and grant GENCI2201). IAGOS gratefully acknowledges the European Commission for the support to the MOZAIC project (1994-2003) the
preparatory phase of IAGOS (2005-2013) and IGAS (2013-2016), the partner institutions of the IAGOS Research Infrastructure : FZJ, DLR, MPI, KIT in Germany, CNRS, Météo-France, Université Paul Sabatier in France and University of Manchester in United Kingdom) and the participating airlines (Lufthansa, Air France, China Airlines, Iberia, Cathay Pacific) for the transport free of charge of the instrumentation. The MOZAIC-IAGOS database is hosted by French Atmospheric Data Center AERIS.



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





**Appendix A – IASI validation and INCA evalution**

Table A1. Ozonesonde stations used for the IASI O$_3$ validation. "N days" represents the number of measurements matching the coincidence criteria.

| Midlatitudes | | | | | | |
|---|---|---|---|---|---|---|
| Station | Location | | N days | Station | Location | | N days |
| Ankara | 39.97°N | 32.86°E | 17 | Kelowna | 49.93°N | 119.40°W | 55 |
| Aquila | 42.38°N | 13.31°E | 7 | Lauder | 45.04°S | 169.68°E | 6 |
| Barajas | 40.47°N | 3.58°W | 65 | Legionowo | 52.40°N | 20.97°E | 108 |
| Bratts Lake | 50.20°N | 104.70°W | 55 | Lindenberg | 52.21°N | 14.12°E | 121 |
| Broadmeadows | 37.69°S | 144.94°E | 14 | Macquarie Island | 54.50°S | 158.94°E | 59 |
| Churchill | 58.74°N | 94.07°W | 7 | Payerne | 46.49°N | 6.57°E | 300 |
| De Bilt | 52.10°N | 5.18°E | 92 | Praha | 50.01°N | 14.45°E | 120 |
| Edmonton | 53.55°N | 114.11°W | 1 | Stony Plain | 53.55°N | 114.11°W | 19 |
| Egbert | 44.23°N | 79.78°W | 64 | Uccle | 50.80°N | 4.35°E | 334 |
| Goose Bay | 53.31°N | 60.36°W | 53 | Ushuaia | 54.85°S | 68.31°W | 26 |
| Hohenpeissenberg | 47.80°N | 11.00°W | 218 | Valentia | 51.93°N | 10.25°W | 100 |
| Huntsville | 34.72°N | 86.64°W | 2 | Wallops Island | 37.90°N | 75.70°W | 6 |
| Tropics | | | | | | | |
| Station | Location | | N days | Station | Location | | N days |
| ascension | 7.97°S | 14.40°W | 24 | Natal | 5.49°S | 35.80°W | 25 |
| Hanoi | 21.02°N | 105.80°E | 9 | Paramaribo | 5.81°N | 55.21°W | 1 |
| Hilo | 19.43°N | 155.04°W | 11 | Reunion | 21.06°S | 55.48°E | 42 |
| Hong Kong | 22.31°N | 114.17°E | 39 | San Cristobal | 0.92°S | 89.60°W | 7 |
| Irene | 25.90°S | 28.22°E | 1 | Santa Cruz | 28.46°N | 16.26°W | 3 |
| Java | 7.50°S | 112.60°E | 1 | Watukosek | 7.50°S | 112.60°E | 1 |
| Nairobi | 1.27°S | 36.80°E | 3 | | | | |
| High latitudes | | | | | | | |
| Station | Location | | N days | Station | Location | | N days |
| alert | 82.50°N | 62.34°W | 77 | marambio | 64.24°S | 56.62°W | 81 |
| belgrano | 77.85°S | 34.55_W | 1 | neumayer | 70.67°S | 8.27°W | 161 |
| Davis | 68.58°S | 77.97°E | 28 | Ny Ålesund | 78.92°N | 11.92°E | 30 |
| eureka | 80.05°N | 86.42°W | 107 | Resolute | 74.72°N | 94.98°W | 87 |
| jokioinen | 60.82°N | 23.50°E | 6 | Scoresbysund | 70.48°N | 21.95°W | 21 |
| keflavik | 63.97°N | 22.60°W | 5 | Sodankylä | 67.36°N | 26.63°E | 35 |





| lerwick | 60.13°N | 1.18°W | 143 | Syowa | 69.00°S | 39.58°E | 67 |
| McMurdo | 77.85°S | 166.67°E | 2 | Thule | 76.56°N | 68.77°W | 10 |
| maitri | 70.46°S | 11.45°E | 5 | | | | |

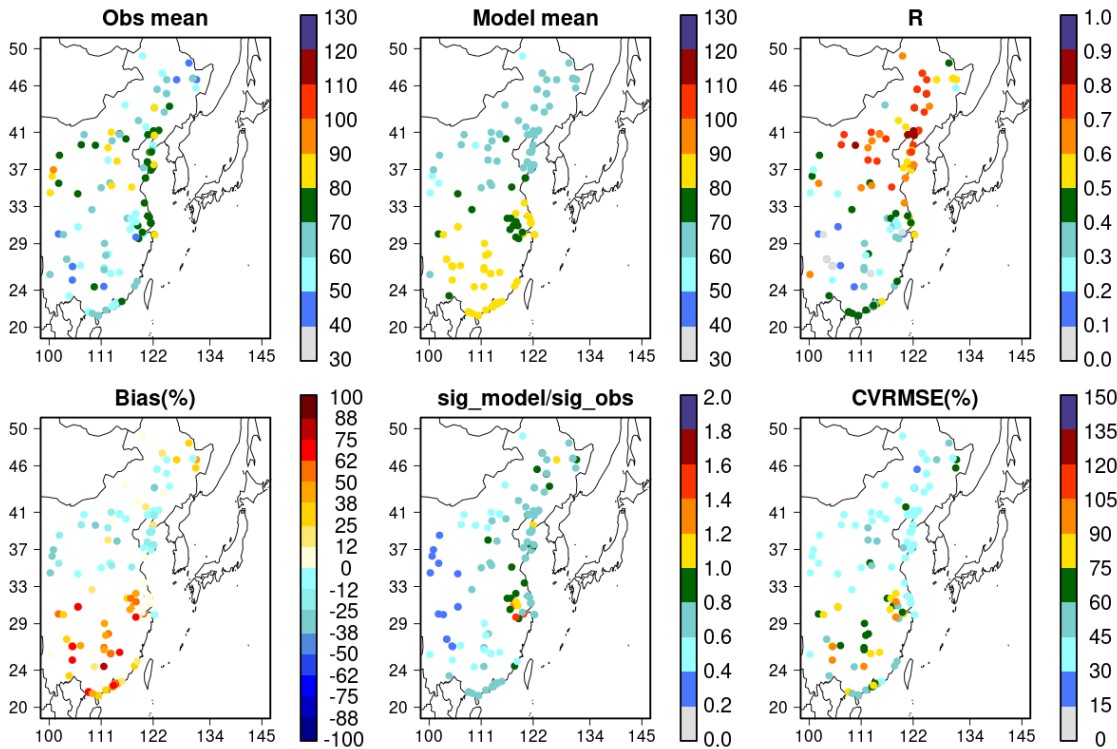

5  Figure A1: Mean daily $O_3$ concentrations (in µg/m3) observed at the rural-type Chinese stations and simulated by INCA for 2014-2017. The statistics of the comparison at each station is shown with the correlation coefficient (R), the mean bias, the ratio of the standard deviation of the model and the observations, and the coefficient of variation of the Root Mean Square Error (CVRMSE).



Figure A2: Mean O₃ partial columns for 2008-2017 observed with IASI (left panels), simulated by INCA (middle panels), and their differences (right panels). The INCA ozone profiles are smoothed by each individual IASI AKs. Four different partial columns are considered: 0-3km, 3-6km, 6-9km, and 9-12km columns.



## Appendix B – Trend analyses

Table B1: Lower free tropospheric $O_3$ trends (3-6km column) calculated from IASI and INCA for the CEC for different time periods.

|  | IASI | INCA |
|---|---|---|
| 2008-2017 | -0.07 ± 0.02 (p<0.01) | -0.08 ± 0.02 (p<0.01) |
| 2009-2017 | -0.08 ± 0.02 (p<0.01) | -0.08 ± 0.03 (p<0.01) |
| 2010-2017 | -0.08 ± 0.02 (p<0.01) | -0.06 ± 0.03 (p<0.01) |
| 2008-2016 | -0.07 ± 0.02 (p<0.01) | -0.09 ± 0.03 (p<0.01) |
| 2008-2015 | -0.06 ± 0.02 (p<0.01) | -0.10 ± 0.03 (p<0.01) |
| 2009-2016 | -0.08 ± 0.03 (p<0.01) | -0.09 ± 0.03 (p<0.01) |

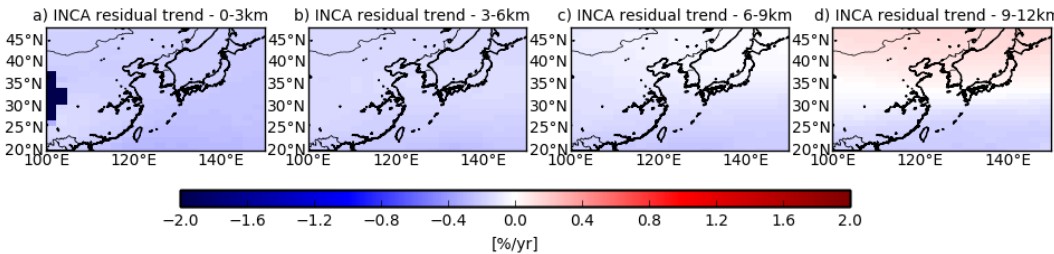

Figure B1: Residual $O_3$ trend derived from the INCA simulation in which all the emissions, $CH_4$ and meteorology are kept constant to their 2007 values. The trend is calculated for the 0-3km, 3-6km, 6-9km, and 9-12km $O_3$ partial columns.



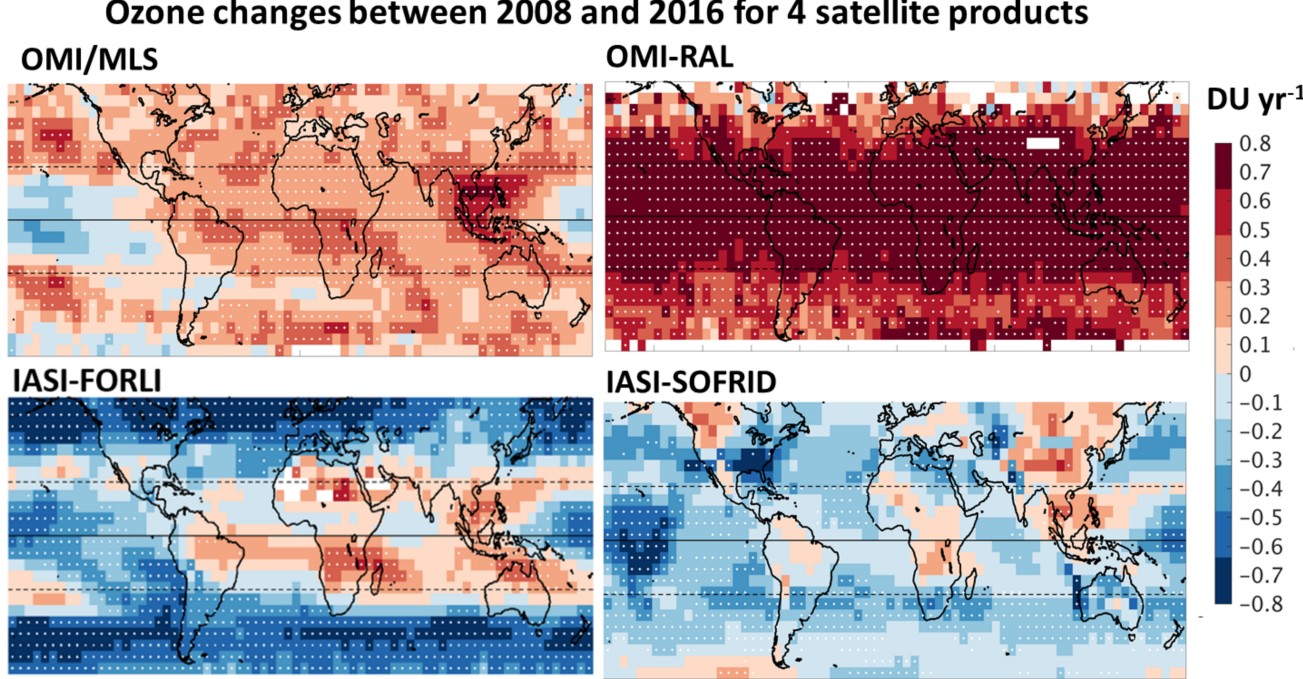

Figure B2: Global gridded ozone trends between 2008 and 2016 derived for the tropospheric columns from four satellite products (OMI/MLS, OMI-RAL, IASI-FORLI, and IASI-SOFRID) in the TOAR-I framework. The figure is an extension/update of Fig. 24 from Gaudel et al. (2018), under the CC-BY 4.0 copyright.