# Peer review of "Recent ozone trends in the Chinese free troposphere: role of the local emission reductions and meteorology"

_Atmospheric Chemistry and Physics, 2021_

## Author Response (AR1)

The authors thank the referees for their interest in the manuscript. Their suggestions, recommendations and remarks were very useful for improving the manuscript. In the following referees' comments are indicated in italics and grey color and the reply to each comment is given just below. The corresponding changes in the manuscript are in blue.

**Referee #1**

*This paper addresses important issues for understanding the evolution of ozone pollution over China and how it is responding to air quality regulations. This is a complicated problem so although this study does not resolve discrepancies between data sets, it is a careful analysis and represents an advancement in our understanding of Chinese ozone pollution. I recommend publication once minor issues are addressed.*

*Specific comments:*

*p5 line 8: I could not find references at https://quotsoft.net/air/ regarding the accuracy and precision of this data. Please provide more information and/or other references that use the data.*

The surface dataset, we use in this study, is the dataset from the Chinese air quality network provided by the China National Environmental Monitoring Centre (CNEMC) of the MEE and archived at this web address. It has been used in several other studies. This is now mentioned in the text and references to two recent studies are done (Li et al., ACP, 2020 and Yin et al., ACP, 2021).

Observational data are issued from the China National Environmental Monitoring Centre (CNEMC) and archived at https://quotsoft.net/air/ (last access: 3 May 2021). The dataset provides hourly data of criteria pollutants $SO_2$, $O_3$, $NO_2$, $CO$, $PM_{2.5}$ and $PM_{10}$ consolidated every day in near real time from May 2014. It has been used in several other studies (e.g. Li et al., 2020; Yin et al., 2021).

*p7 line 1: Zheng et al., 2018 uses the MEIC inventory – I assume that also used here? This should be stated explicitly since there are other bottom-up inventories for China, e.g., as mentioned in Zheng, B., et al., (2018), Rapid decline in carbon monoxide emissions and export from East Asia between years 2005 and 2016, Environ. Res. Lett., 13(4), 044007, doi:10.1088/1748-9326/aab2b3.*

Indeed, we use the MEIC inventory. This is now stated in the text.

For China, the anthropogenic emission inventories are replaced the MEIC (Multi-resolution Emission Inventory for China) inventory (Zheng et al., 2018) available for the period 2010-2017.

*p8 line 12: Could the lower bias and degraded correlation with IASI-IAGOS compared to IASI-ozonesondes also be related to the assumption of a vertical profile at the lat/lon of the mid-point in a slanted profile? This might explain the low bias in the lowest layer since those measurements would be taken closer to urban areas near the airport.*

The degraded statistics when comparing IASI to IAGOS are due to a combination of factors, difficult to identify and quantify. We agree that the assumption of a vertical profile instead of a slant profile is likely an additional factor. We added these sentences in the text "In addition, the comparison I made assuming a vertical profile at the latitude and longitude of the mid-point of a slant profile recorded during take-off and landing phases. The lowest part of the profile may be more influenced by urban areas near the airport and then less reproduced by IASI due to its limited sensitivity close to the surface."

*Fig. A1: In addition to the 2014-2017 mean and statistics, it would be useful to see the trends in ground station observations compared to the model surface ozone.*

The 2014-2017 period is short to be able to calculate robust trends. Most of the trends calculated at the surface from surface measurements and model simulations has a p-value larger than 0.05. It is then difficult to conclude. Below is the map of the observed and simulated trends at the station location when p-values are both smaller than 0.05. The trends derived from the model are rather consistent but slightly smaller than the trends derived from the surface measurements. The stations

for which a strong disagreement appears correspond to the ones where the model failed to correctly reproduce ozone concentrations. This figure will be added in the appendix and these sentences have been added in the text "In terms of trends, the 2014-2017 time period is short to derive robust trends. When considering observed and simulated trends at the surface with p-values smaller than 0.05, model and observations are rather consistent, especially in regions where the model simulates correctly the $O_3$ concentrations (Fig. A2). The model tends to underestimate the positive trends compared to the observations."

[Figure]

*Wording change suggestions:*
*p2 line 15: "have been applying" to "have been enacted"*
This has been corrected.
*p3 line 3: "In this study, we question the ability..." to "In this study, we examine the ability..."*
This has been corrected.
*p3 line 8: "Results are also discussed in light with the TOAR outcomes" to "Results are also contrasted with the Gaudel et al., (2018) TOAR outcomes"*
This has been corrected.
*p4 line 28: Since validation references are not given in this section, I suggest adding a line: "Initial validation of the KOPRAFIT IASI ozone retrievals with ozonesonde and IAGOS data is presented in Section 3."*
This has been added
*p8 line 14: "worser" to "lower"*
This has been corrected.
*p9 line 23: "less than 300 data" to "less than 300 profiles"?*
It corresponds to the number of measurement points and not profiles. We corrected it to "less than 300 measurement points" to make it clearer for the reader.
*p11 line 19: "desertic" to "desert"*
This has been corrected.
*p11 line 21: "This region should not be considered here." to "This region is not considered here."*
This has been corrected.
*p11 line 24: "This translates even stronger to the model" Not quite sure what is meant by this– maybe "This feature becomes even stronger in the model when AKs are applied"?*
We've moved to the referee's formulation.
*p17 line 2: "nuding" to "nudging"*
This has been corrected.
*p19 line 20: "These results seem to comfort the consistency" to "These results corroborate the consistency"*
This has been corrected.
*p19 line 25: "and the caution to take to not overinterpret the results" to "and the need for caution to prevent overinterpreting the results.*
This has been corrected.
*p20 line 12: "Some individual studies exist but once again they do not allow one to conclude"*
*Conclude what?*

We have added "to conclude concerning the role of the drifts in the trend discrepancies"
*p21 line 23: "leads to reduce" to "further reduces"*
This has been corrected.

***Referee #2***
*This paper reports ozone trends in the free troposphere over China using the IASI satellite instrument and interprets them using an atmospheric transport model. I have a few comments that the authors should consider when they revise their manuscript. One major comment is associated with the authors not using averaging kernels when they compare their model to the IASI data.*

*Second paragraph: this area of the world is attracting a lot of attention. Already since this paper was submitted there are a few more papers that need citing in the introduction, e.g. https://agupubs.onlinelibrary.wiley.com/doi/10.1029/2021GL092816?af=R*
Reference to this newly paper has been added. As it is difficult to be exhaustive citing all the studies focusing on Chinese ozone pollution, we have also added "e.g." before the references.

*Page 4, Line 17: Three different a priori and constraints....ambiguous.*
We have rephrased this sentence as following hopping it is clearer now. "The a priori and the constraints are different depending on the tropopause height, which is based on the 2 PV geopotential height product from the ECMWF (European Centre for Medium-range Weather Forecasts). One a priori and one constraint is used for polar situations (i.e. tropopause < 10 km), one for midlatitude situations (i.e. tropopause within 10-14 km), and one for tropical situations (i.e. tropopause > 14 km)."

*American spelling: use of centre for ECMWF and elsewhere. Vapor instead of vapour.*
This has been corrected.

*Section 2.3. There is little in the way of data description for the surface measurements. What is this website? How are the measurements collected? Uncertainties? What are criteria pollutants?*
The surface dataset, we use in this study, is the dataset from the Chinese air quality network provided by the China National Environmental Monitoring Centre (CNEMC) of the MEE and archived at this web address. It has been used in several other studies. This is now mentioned in the text and references to two recent studies are done (Li et al., ACP, 2020 and Yin et al., ACP, 2021).
Observational data are issued from the China National Environmental Monitoring Centre (CNEMC) and archived at https://quotsoft.net/air/ (last access: 3 May 2021). The dataset provides hourly data of criteria pollutants $SO_2$, $O_3$, $NO_2$, CO, $PM_{2.5}$ and $PM_{10}$ consolidated every day in near real time from May 2014. It has been used in several other studies (e.g. Li et al., 2020; Yin et al., 2021).

*Page 5, Line 16: ...the larger the amplitude of the diurnal ozone cycle... This is not immediately obvious to me. Surely the boundary layer diurnal variation has a large role to play? This reviewer would have used a short-lived gas or particulate that would be relevant to the distance from anthropogenic activity.*
The method applied is the one developed by Flemming et al. (2005) as referred to in the manuscript. We have rephrased this sentence as follow: "with the assumption that the larger the amplitude of the diurnal ozone cycle is, the more the station is in an urban environment."

*Section 3.1. Here, I would focus on providing the validation that is pertinent to your study. A more detailed validation is always available in another paper.*

As we use a new version of our IASI product for which a detailed validation is not yet available, we provide a short global validation using ozonesondes. Unfortunately, only few ozonesondes are available in East Asia region and their launch time is not compatible with the time coincidence criterion. Then, we use the IAGOS profiles available in this region to propose an evaluation of the IASI retrievals in China. However, validation of satellite product with IAGOS is more difficult as the profiles are limited by the flight cruise altitude of the aircraft.

*Minor point: page 7, line 9: shorter than +/- 6 hours.*
This has been corrected.

*Page 9, line 2: We select IAGOS profiles with top measurements not lower than 500 hPa? This statement is unclear.*
We guess the referee statement referred to page 8 and not 9.
We have changed the part of the text from "We use coincidence criteria…" to "to apply the averaging kernels" to "IAGOS profiles are filtered to coincidence in space and time with IASI pixels using the same criteria as for the ozonesondes (1° around the station, +/-6 hours). The reference point for IAGOS to apply the coincidence criteria is the middle of the profile. IAGOS profiles with data missing above 500hPa do not extend enough in altitude to be correctly compared to IASI observations. Then, they are not analyzed. After these two filters, we count 213 IAGOS profiles for the time period 2011-2017. These selected profiles are extended vertically between the top (usually the cruise altitude of the aircraft) and 60 km with the a priori profiles used in the IASI retrieval. We apply the IASI averaging kernels to these extended IAGOS profiles. The 500 hPa criterion to filter the IAGOS profiles is a compromise to have a reasonable number of profiles to compare with and to limit the contribution of a priori in the lower tropospheric part of the smoothed profiles. Despite this, the resulting smoothed IAGOS profile may still be significantly affected by the a priori profile used to extend the raw IAGOS profile especially when the cruise altitude of the profile is low. "

*Page 9, line 7: most sensitive to what?*
Again, we guess the referee statement referred to page 8 and not 9. We have changed to "most sensitive to ozone".

*Page 9, line 12: why does the column degrade re correlation? It is reverting to the prior?*
Again, we guess the referee statement referred to page 8 and not 9.
The degraded statistics when comparing IASI to IAGOS are due to a combination of factors, difficult to identify and quantify. In addition to the diurnal cycle and the "large" time coincidence considered already mentioned in the text, we also now discuss the assumption of a vertical profile instead of a slant profile a potential factor as well as the limited sensitivity of the retrieval towards the surface. We added these sentences in the text "In addition, the comparison I made assuming a vertical profile at the latitude and longitude of the mid-point of a slant profile recorded during take-off and landing phases. The lowest part of the profile may be more influenced by urban areas near the airport and then less reproduced by IASI due to its limited sensitivity close to the surface."

*Page 9, line 15: last sentence is unclear.*
The sentence has been changed to "The evaluation of IASI using IAGOS is then difficult for the 6-9km column with so few coincident observations."

*Page 9, line 21. Model has a positive bias of 12% compared with the surface stations over China?*
The mean normalized biased of the model compared with the surface stations is 12% all over China, but as it is discussed in the text and shown in Fig. A1, the bias station by station is much more variable.

*Page 11, line 11. Here and elsewhere I strongly disagree with the absence of averaging kernels when the model is compared to the satellite data. It seems as though the default for this study is to ignore*

*them and place the comparisons that do use them in an appendix. It should be the other way around, if anything. You cannot legitimately compare the model with IASI without taking into account the vertical sensitivity of IASI to changes in ozone.*

We agree with the referee that the correct way to compare satellite observations with limited vertical sensitivity with models is to consider the averaging kernels of the observations. We first made the choice to present the comparison between unsmoothed model and IASI as the trends are from the native simulation of the model to remain entirely independent of the observations. We took care of also presenting the results of the comparison with the model smoothed with the IASI AKs and made the choice to present the figure in the appendix not to overload the text of the manuscript with large figures. It is also worth noting that it is also recommended to always consider both raw and smoothed validation datasets in satellite validation procedure to identify strong deviations of satellite data to the atmospheric state, strong contribution of the prior and then to avoid misinterpretation of the validation results. In our case, as our first presentation approach may mislead the readers on the importance of using the averaging kernels when comparing satellites and models, we have moved the former figure A3 in the main text and the former figure 3 in the appendix. We have also rearranged the text p11, lines 2-27 as follow:

"We compare IASI and INCA $O_3$ partial columns over the East Asia domain (100-145°E, 20-48°N) averaged over the 2008-2017 period. To properly compare IASI and INCA, the model is smoothed by IASI averaging kernels (AK) (Fig. 3). The comparison without any smoothing of the model is also shown in Appendix (Fig. A3). This latter comparison is also interesting to determine the deviation of the observations compared to the native model resolution as the simulated trends are calculated without applying the AK to the model to keep them independent of the observations and a priori information used in the retrievals (see Section 4). Note that Barret et al. (2020) also discussed the interest of considering raw and smoothed data in satellite data procedures. Spatial distribution and spatial gradients of ozone are in good agreement for the 4 partial columns. On average, the differences are smaller than 5% for the 0-3km and 3-6km columns. The difference is larger for the upper columns - 6-9km and 9-12km columns – with a mean negative difference of about 7% for both 6-9km and 9-12km columns. For the 0-3km partial column, it is worth noting that the IASI retrieval is not highly sensitive to these altitudes and that the a priori contribution is larger (Dufour et al., 2012). This is illustrated over China by IASI systematically smaller than INCA without AK smoothing (-5% to -25%, Fig. A3a) and an improved agreement ( +/-5%) when applying the AK to the model (Fig. 3a) and by larger differences over tropical maritime regions reduced when AK are applied. The agreement between IASI and INCA remains largely reasonable accounting for the observation and model uncertainties. For the 3-6km partial columns, where the IASI retrievals are the most sensitive, a very good agreement between IASI and INCA, within +/-10%, is observed for a large part of the domain (Figs 3b and A3b). It is the partial columns for which the agreement is the best. For the upper columns (6-9km and 9-12km), IASI is almost systematically smaller than INCA over the domain (Figs 3c-d and A3c-d). IASI is always smaller than INCA over the most part of China whatever the partial columns considered. IASI is mainly larger than INCA in the lower troposphere and smaller in the upper troposphere elsewhere. In the desert northwestern part of the domain, even if the emissivity is included in the IASI retrievals, the quality of the retrievals can be affected and confidence in the data reduced. This region is then not considered here. The retrieval in the tropical-type airmasses have been shown to reinforce the natural S-shape of the ozone profiles, leading to some overestimations of ozone in the lower troposphere and an underestimation in upper troposphere (Dufour et al., 2012). This likely explains the positive and negative differences with the model in the southeastern part of the domain (Fig. A3). Globally, the differences between IASI and INCA are the smallest over the Central East China (CEC). Figure 4 shows the IASI and INCA monthly timeseries of the different $O_3$ partial columns between 2008 and 2017 for this region. The INCA timeseries are the series from which the trends are derived (Section 4) and then do not include any smoothing from the IASI AKs."

*Page 13, line 11: confirm this corresponds to two sigma. Explain to the readers why you've opted for that measure.*

We use an ordinary least-square regression to determine the slope and then the trend. In that case, the calculation of the uncertainty on the slope should be based on the t-test (95% confidence) and calculated as t*sigma. In our case the number of points used in the regression is large enough and the uncertainty corresponds roughly to two sigma. We rephrase the corresponding sentence as follow: "As an ordinary linear regression is used for trend calculation, the trends uncertainties are calculated as the t-test value multiplied with the standard error of the trends, which correspond to the 95% confidence interval."

*Page 16, line 16: typo.*
It has been corrected

*Page 17, line 2: typo or at least I think so!*
It has been corrected!

*Table 4: the sensitivity calculations appear to be a progressive (cumulative) degradation of the reference case. In that case, won't the sensitivity calculations become more non- linearly different from the reference case?*

As explained in the text, the objective of these simulations is to remove one-by-one the interannual variability and trend induced by the different processes (different emission source and meteorology). This approach allows us to determine the residual variation of ozone due to the fact that the INCA model is a nudged GCM (u and v wind components solely are nudged) and then some variables as temperature, convection, etc are recalculated internally during the simulations. Then, after removing the residual ozone variations to the different simulations, we calculate the trends induced by the different processes and compared them to the total trend to determine their relative contribution to the trend. We have checked that the sum of the individual trends components taken separately is equal to the total trend from the reference simulation. This ensures that the procedure adopted to isolate the individual contributions to the trend does not affect the conclusions due to possible non-linearities. Thanks to referee comment, we have realized that the notations used in Table 4 were misleading. We have changed the table in the revised version of the manuscript as follow:

**Table 4: Description of the different simulations done with the INCA model and the trends calculation. Note that for all the simulations the biogenic emissions are constant over the period.**

| Simulations[a] | Trend calculation[b] | |
| --- | --- | --- |
| SR = Reference | Total | trend(SR) |
| SC = no China variations | China | trend(SR) – trend(SC) |
| SG = no China & no global variations | Global | trend(SC) – trend(SG) |
| SB = no China & no global & no biomass burning variations | BBg | trend(SG) – trend(SB) |
| S4 = no China & no global & no biomass burning & no methane variations | CH4 | trend(SB) – trend(S4) |
| SM= no China & no global & no biomass burning & no methane & no meteo variations | MET | trend(S4) |

[a] all the simulations are corrected from residual O$_3$ variations (see text)
[b] the contribution of each process to the trend is calculated by dividing the corresponding trend by the total trend (TT).

We also present now the trend induced by each process in Fig 6 in addition to their contribution to the trend as well as the total trend. The new Fig. 6 focuses on 3-6 km columns. The figure for the 6-9 km columns is reported in the appendix. In addition, the last column of former Fig. 5 presenting the trend excluding the residual has been removed as the information is now in Fig. 6 .

*Page 18: while discussing the impact of various sensitivity calculations it would be useful to link them back to the model and observed comparisons. For instance, the influence of the biomass burning in the export region looks impressive in the model (Figure 6) but the model doesn't look particularly favourable against the data in Figure 5. BTW, Figure 5 appears to be cropped.*

As we have modified Fig. 6 in the revised manuscript, we also revised the text and includes a discussion of large contribution of processes such as biomass burning as follow:

"The contribution of the different processes is shown in Fig. 6 for the 3-6km columns and in Fig. B2 for the 6-9km columns. We focus on these two columns as it corresponds to the tropospheric regions where IASI is the most sensitive and then where agreement with the model is the best. We also focus mainly on China where the p-values of the total trend are smaller than 0.05. In the lower free troposphere, the main contributions to the trends are the local Chinese emissions and the meteorology, with contributions larger than 20%. The other tested variables (global emissions, biomass burning emissions and methane) contribute to the trends within 20% over mainland China, with a negative contribution of methane for the entire domain. This means that the increase in methane concentrations and then the associated ozone production counteracts the ozone reduction due to the other processes (emissions and meteorology). In the upper free troposphere, meteorology and Chinese emissions also dominate the contributions. Biomass burning emissions play also a more important role to explain the trend in the North China Plain between Beijing region and the Yangtse River. In the south of the domain where the robustness of the trends might be discussed as p-values are larger than 0.05, strong compensations seem to operate between the contributions of the different processes leading to a less robust assessment of their respective contributions."

Concerning the crop in Figure 5, we have fixed that in the revised version of the manuscript.

*Section 5.2 was interesting but I think it would benefit from being more accurate in describing differences between TOAR and this study, even if the authors run the risk of repeating themselves, e.g. differences in time period.*

In the head of section 5.2, we have added the following information: "It is worth noting that this study brings another angle to the question of ozone changes with time over China than previously discussed during Phase I of TOAR. In the TOAR effort, our IASI product was used to describe the seasonal variability of ozone for the time period 2010-2014 over China. However, this product was not used for assessing ozone trends. Compared to the other IASI products, our product is more sensitive to the lower part of the troposphere (see Table 2 in Gaudel et al., 2018). We hope this study can bring more information to fuel the discussion on the discrepancies between the UV and IR techniques."

*Page 19: length of period. Sensitivity to perturbations can be removed by using, for example, the Theil-Sen estimator. Using that approach may also reduce the sensitivity of your results to end points.*

We have recalculated trends of table B1 using the Theil-Sen estimator. As we rounded the values at 1e-2, all the trends derived from IASI remain unchanged. For the model, all the calculated trends are increased by 0.01, except for 2008-2017, which is unchanged.

We have added this sentence at the end of the discussion "It is worth noting that using the Theil-Sen estimator to calculate the trends changes only slightly the trends."

*Page 21: impact of sampling. To check consistency would it be useful for the authors to sample the model coincident with IASI and then again with IAGOS and compare the trends using the two sets of sampled data?*

We have not analyzed the impact of the temporal sampling independently of the spatial sampling as suggested by the referee. In our case, we consider simultaneously the temporal and the spatial coincidence of the IASI and INCA model to IAGOS profiles. However, if Fig 5 we compare the simulated trends sampled or not at the IASI pixels for China. They are not significantly affected by the sampling. For Europe, which is discussed here, the trends derived from INCA and IASI for the tropospheric column for a latitude band 35-48°N close to the one where model/observation agreement is the best in China are consistent (-0.24 DU/yr and -0.14 DU/yr, respectively).

*For Table 6, I suggest also include the number of points in each calculation since this value will change a lot for the rows and columns.*

The number of profiles has been included in Table 6.